# The Air-temperature Response to Green/blue-infrastructure Evaluation Tool (TARGET v1.0): an efficient and user-friendly model of city cooling.

Ashley M. Broadbent[1,2,3,4], Andrew M. Coutts[3,4], Kerry A. Nice[3,4,5], Matthias Demuzere[6,7], E. Scott Krayenhoff[8,1,2], Nigel J. Tapper[3,4], and Hendrik Wouters[7,6]

[1]School of Geographical Sciences and Urban Planning, Arizona State University, Tempe, Arizona, USA
[2]Urban Climate Research Center, Arizona State University, Tempe, Arizona, USA
[3]School of Earth, Atmosphere and Environment, Monash University, Clayton, Australia
[4]Cooperative Research Centre for Water Sensitive Cities, Melbourne, Australia
[5]Transport, Health, and Urban Design Hub, Faculty of Architecture, Building, and Planning, University of Melbourne, Victoria, Australia
[6]Ghent University, Laboratory of Hydrology and Water Management, Ghent, Belgium
[7]KU Leuven, Department of Earth and Environmental Sciences, Celestijnenlaan, Leuven, Belgium
[8]School of Environmental Sciences, University of Guelph, Guelph, Ontario, Canada.

*Correspondence to:* Ashley M. Broadbent (ashley.broadbent@asu.edu)

**Abstract.**

The adverse impacts of urban heat and global climate change are leading policy-makers to consider green and blue infrastructure (GBI) for heat mitigation benefits. Though many models exist to evaluate the cooling impacts of GBI, their complexity and computational demand leaves most of them largely inaccessible to those without specialist expertise and computing facilities.

Here a new model called **T**he **A**ir-temperature **R**esponse to **G**reen/blue-infrastructure **E**valuation **T**ool (TARGET) is presented. TARGET is designed to be efficient and easy to use, with fewer user-defined parameters and less model input data required than other urban climate models. TARGET can be used to model average street level air temperature at canyon-to-block-scales (e.g. 100 m resolution), meaning it can be used to assess temperature impacts of suburb-to-city-scale GBI proposals. The model aims to balance realistic representation of physical processes and computation efficiency. An evaluation against two different datasets shows that TARGET can reproduce the magnitude and patterns of both air temperature and surface temperature within suburban environments. To demonstrate the utility of the model for planners and policy-makers, the results from two precinct-scale heat mitigation scenarios are presented. TARGET is available to the public and ongoing development, including a graphical user interface, is planned for future work.

## 1 Introduction

Policy and decision-makers are increasingly aware of the cooling potential of green and blue infrastructure (GBI). Recent examples of this include the Australian Federal Government's 20 Millon Trees Program (Commonwealth of Australia, 2017) and Singapore Green Plan (Singapore Ministry of Environment and Water Resources, 2012). Governments and urban planners wish to evaluate the cooling effects of design and planning options. Urban climate models are becoming more complex, allowing more complete representation of urban physics. However, the complexity of urban climate models renders them inaccessible to consultants (Elasson, 2000), who typically work for designers and planners. Commonly used urban climate models, such as the Weather Research and Forecasting (WRF) model (Skamarock et al., 2008) and ENVI-met (Bruse, 1999) require trained research scientists and significant computational resources to run. As a result, consultants usually provide generic and unsubstantiated estimates of cooling magnitudes. Consequently, there is a need for simple, computationally efficient, and scientifically-defensible urban climate models that can be used by consultants to provide reliable estimates of cooling to governmental planners and policy-makers.

There are a number of existing micro-to-local scale urban models capable of modelling GBI. The most complex models are primary based on computational fluid dynamics (CFD) techniques. These include ENVI-met, hand-coded CFD models (such as OpenFOAM (OpenFOAM, 2011) or STAR-CD (CD-adapco, 2011)), and other CFD based approaches (Bailey et al., 2014, 2016; Kunz et al., 2000; Schlünzen et al., 2011; Yamada and Koike, 2011; Bruse, 1999). ENVI-met is the most commonly used urban microclimate model. However, numerous ENVI-met studies have reported concerns with model accuracy, particularly for representation of vegetation (Ali-Toudert and Mayer, 2006; Krüger et al., 2011; Acero and Herranz-Pascual, 2015; Spangenberg and Shinzato, 2008). In addition, the complexity of configuration and computational intensity of all CFD based models (i.e. 24 hours of simulation requiring 24 hours of computation time) puts their usage out of the reach of non-specialized users.

A second group of commonly used models, such SOLWEIG (Lindberg et al., 2008) and RayMan (Matzarakis et al., 2007, 2010), focus around radiation fluxes in urban areas. These models have been used to assess GBI cooling, especially tree shading. However, the limitations of these models may not allow a complete assessment of GBI cooling because the effects of evapotranspiration are neglected. The Temperatures of Urban Facets in 3D (TUF-3D) model (Krayenhoff and Voogt, 2007) and a vegetated derivative (VTUF-3D) (Nice et al., 2018), provide a precise representation of urban canyon physical processes. However, TUF-3D and VTUF-3D require a high level of computer power, modelling experience, and parameter setup.

The canyon air temperature (CAT) model (Erell and Williamson, 2006) shows potential as a computationally efficient model that calculates air temperatures using urban building and vegetation geometry and moisture availability. However, the lack of surface temperature prediction makes it difficult to derive human thermal comfort indexes. The Town Energy Balance (TEB) model (Masson, 2000) has emerged as a popular urban area parameteristion scheme. The TEB-Veg (Lemonsu et al., 2012; Redon et al., 2017) variation includes urban vegetation and provides functionality to assess cooling impacts of GBI. However, the TEB-Veg model configuration and application requires a level of modelling skill normally outside the capability of environmental consultants.

While not an air temperature model, the Local-Scale Urban Meteorological Parameterization Scheme (LUMPS) (Grimmond and Oke, 2002) has been widely used to assess the impacts of GBI on surface energy balance (SEB). The Surface Urban Energy and Water Balance Scheme (SUEWS) (Järvi et al., 2011), a superset of LUMPS with added urban water balance functionality, provides a means to assess vegetation (and associated soil) transpiration impacts at local scales. SUEWS shows good performance in SEB evaluations for Vancouver and Los Angeles (Järvi et al., 2011), Helsinki (Järvi et al., 2014), and Singapore (Demuzere et al., 2017). Due to the success and simplicity of LUMPS, we use it as a key component of the model presented here.

The lack of an efficient yet accessible tool for assessing GBI has been identified as a research gap. Here we introduce and evaluate a new model called **T**he **A**ir-temperature **R**esponse to **G**reen/blue-infrastructure **E**valuation **T**ool (TARGET). TARGET is a simple modelling tool that calculates surface temperature and street level (below roof height) air temperature in urban areas. TARGET is designed to make quick and accurate assessments of urban temperatures and GBI cooling impacts with minimal input data requirements. TARGET calculates the average air temperature at street level in urban areas, but does not represent micro-scale variations of radiation exchange or wind flow at the human scale. The model is designed to be used at the urban canyon-to-block scales (100–500 m). We recommend a minimum spatial resolution of 100 m for air temperature simulations and 30 m for surface temperature. It can be used to assess the canyon averaged impacts of street scale interventions or larger-scale suburban greening projects. TARGET is climate-service-oriented tool that provides a first order approximation of the impacts of GBI on surface temperature and street level air temperature to provide scientific guidance to practitioners during the planning process. The computational efficiency of the model is such that a user (with 1–2 hours of training) could calculate in minutes the 100 m horizontal resolution cooling effects, on a normal desktop computer, across an entire suburb/local-government area or neighbourhood.

The main aims of this paper are the following: (1) to provide a technical description of TARGET; (2) to provide detailed evaluation of model performance; and (3) to provide proof of concept, and illustrate how the model could be operationalized by a consultants and practitioners.

## 2 Model description

### 2.1 Model overview

As outlined in Fig. 1, TARGET treats each model grid point as an idealized urban canyon with roofs, walls, and ground-level facets. Roof width ($W_{roof}$), building height ($H$), tree width ($W_{tree}$), and street width ($W$) are used to define the geometry of the canyon. The thermal and radiative characteristics of roofs and walls are considered to be uniform. At street level, the surfaces can be defined as: concrete, asphalt, grass, irrigated grass, and water. Trees are represented at roof height and the

10 surfaces beneath trees are considered to be representative of the ground level surfaces. To represent the first order shading impacts of trees, we effectively represent tree canopy as part of the urban canyon. As shown in Figure 1, the width of the canyon (and therefore the amount of radiation that enters and leaves the canyon) is modulated by the planar area of trees. The simple method, implies that none of the radiation effectively "intercepted" by trees enters the canyon. The area underneath trees (not shown in planar land cover maps) is added to the model to represent the additional thermal mass. This simple approach

allows for a first order representation of two major process associated with trees: solar shading and longwave trapping.

Additionally, water bodies are treated separately to all other surfaces using an independent module. More details about the model process is shown in Fig. 2. For each grid point, the average surface characteristics are used to calculate an aggregated surface temperature ($T_{surf}$). $T_{surf}$ is converted to an average canopy layer air temperature ($T_{ac}$), using a estimated canopy wind speed ($U_{can}$), and above canopy air temperature ($T_b$). A uniform $T_b$ for all grid points is diagnosed for each timestep

using reference meteorological variables.

### 2.2 Input data requirements

#### 2.2.1 Land cover

TARGET uses simple data inputs that are intended to be easily accessible. The model requires the user to define the plan area of buildings ($A_{roof}$), concrete ($A_{conc}$), asphalt ($A_{asph}$), grass ($A_{gras}$), irrigated grass ($A_{igrs}$), tree ($A_{tree}$), and water ($A_{watr}$).

These land cover categories are self-explanatory and describe most of the surfaces present in urban areas. Local governments often have geographical information system (GIS) datasets of land cover and/or land-use that can be used for land cover input data. Further, we intend to develop a graphical user interface (GUI) that allows users to easily input land cover datasets and define the model domain. This feature will allow users to convert and upload GIS data (e.g. shape and raster files) directly into the model. The $W_{roof}$, $W_{tree}$, $W$, $W*$, and wall area ($A_{wall}$) are calculated from plan area land cover inputs. However,

average building height (m) must be user defined or set to a domain average value. If detailed land cover data are not available, input data can be defined from existing land-use look-up tables or from databases such as the World Urban Database and Portal

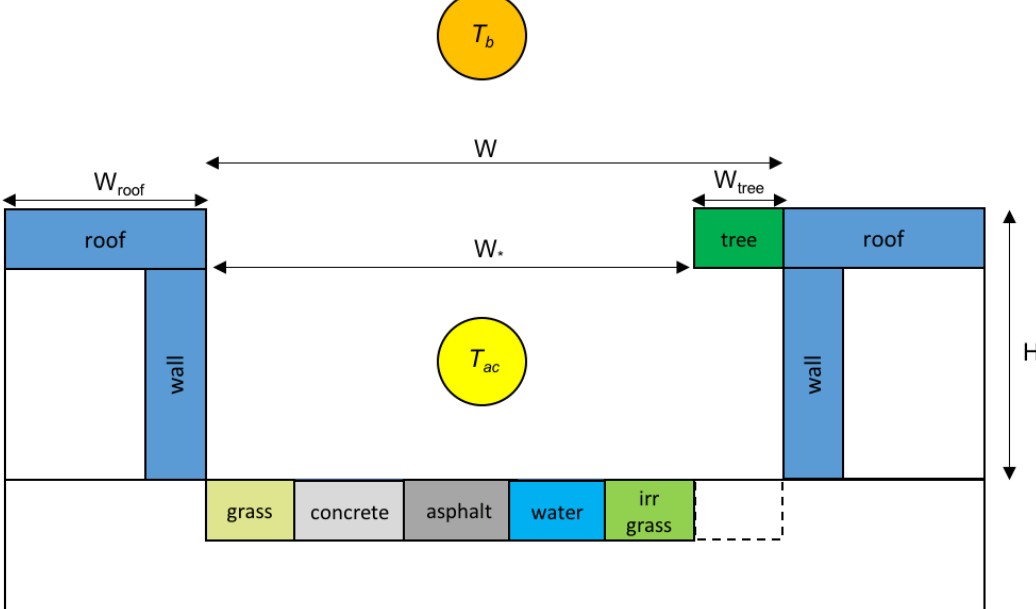

**Figure 1.** Schematic of TARGET urban canyon setup. $T_{ac}$ = canopy layer air temperature and $T_b$ = above canopy air temperature, which is a uniform value across the whole domain. $W_{roof}$ is the roof width, $W_{tree}$ is the tree width, $W$ is canyon width, and $W* = W - W_{tree}$. The surface beneath trees is assumed to be representative of canopy ground-level surfaces.

Tool (WUDAPT) (Mills et al., 2015; Ching et al., 2018). See Wouters et al. (2016) for an example of how the WUDAPT data could be integrated.

### 2.2.2 Meteorological data

TARGET requires reference meteorological data to drive the model and calculate street level air temperature. The following meteorological variables are required: incoming shortwave (solar) radiation ($K\downarrow$), incoming longwave (terrestrial) radiation ($L\downarrow$), relative humidity ($RH$), reference wind speed (typically at 10 m) ($U_z$), and air temperature ($T_a$). The user must define the height above ground of reference $U_z$ and $T_a$. Meteorological data should be representative of a nearby airport or an open site with minimal buildings. At a minimum, reference meteorological data should conform to World Meteorological Organization guidelines (Oke, 2007).

### 2.3 Radiation calculation

The net radiation of the $ith$ surface type ($R_{n,i}$) is calculated using the following:

$$R_{n,i} = \left( K\downarrow (1-\alpha_i) + \epsilon_i \left( L\downarrow - \sigma T^4_{surf,i,[t-2]} \right) \right) SVF_i \tag{1}$$

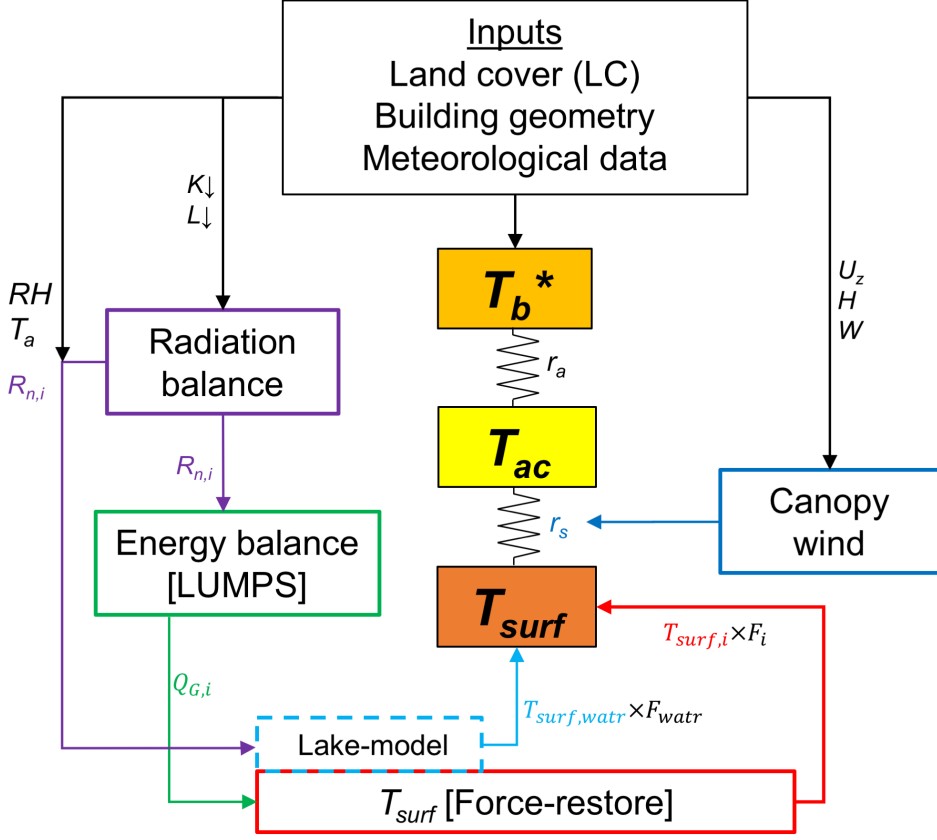

**Figure 2.** Overview of approach used in TARGET . $T_{ac}$ is street level (urban canopy layer) air temperature (°C), $T_b$ is the air temperature above the urban canopy layer (°C), $T_{surf,i}$ is the surface temperature for surface type $i$, $K\downarrow$ is incoming shortwave radiation (W m$^{-2}$), $L\downarrow$ is incoming longwave radiation (W m$^{-2}$), $T_a$ is reference air temperature (°C), $R_n$ is net radiation (W m$^{-2}$), $RH$ is relative humidity (%), $F_i$ is fraction of land cover type $i$ (%), $Q_{H,i}$ is sensible heat flux for surface $i$ from LUMPS (W m$^{-2}$), $Q_{G,i}$ is storage heat flux for surface type $i$ from LUMPS (W m$^{-2}$), $U_z$ is reference wind speed (m s$^{-1}$), $H$ is average building height (m), $W$ is average street width (m), $r_s$ is resistance from surface to canopy (s m$^{-1}$), and $r_a$ is resistance from urban canopy to the atmosphere (s m$^{-1}$). *$T_b$ is a homogeneous value for the whole domain, which is diagnosed by the processes laid out in Sect. 2.7.

where $\alpha_i$ is surface albedo, $\epsilon_i$ is surface emissivity, and $\sigma$ is the Stefan-Boltzmann constant ($5.67 \times 10^{-8}$ W m$^{-2}$K$^{-4}$). The $\alpha_i$ and $\epsilon_i$ values are predefined for each surface (see Table 1). The right hand side of the equation accounts for net longwave radiation. The modelled $T_{surf,i,[t-2]}$ from 2 time steps $(t)$ previously is used to calculate $L\uparrow$. This is necessary to avoid circular logic in model calculations; modelled $T_{surf,i,[t-2]}$ is calculated using the storage heat flux $(Q_{G,i})$, which takes $R_n$ from the previous time step. The time lag does not significantly affect calculations when a 30 minute time step is used. The average sky view factor $(SVF_i)$ is included to broadly represent the interception of incoming and outgoing short and longwave radiation by buildings and trees on the radiation balance. Addition of $SVF$ restricts the net radiation exchange of each facet to its total view factor occupied by sky. It assumes that walls and ground surfaces have similar longwave emission relative to the sky, and that solar radiation receipt can be approximated by $SVF$, on average. This simplification means that the model makes no distinction between lit and unlit buildings walls and roads. $SVF_i$ for ground-level, wall, and roof facets is defined as (Sparrow and Cess, 1978):

$$SVF_{ground} = \left[1 + \left(\frac{H}{W*}\right)^2\right]^{\frac{1}{2}} - \frac{H}{W*}; \tag{2}$$

$$SVF_{wall} = \frac{1}{2}\left(1 + \frac{W*}{H} - \left[1 + \left(\frac{W*}{H}\right)^2\right]^{\frac{1}{2}}\right); \tag{3}$$

$$SVF_{roof} = 1 \tag{4}$$

The $R_{n,i}$ is then used to calculate a $Q_{G,i}$ for each surface type.

## 2.4 Storage heat flux ($Q_G$) calculation

The storage heat flux $(Q_{G,i})$ for the $ith$ land cover class is calculated using an adapted version of the objective hysteresis model (OHM) (Grimmond and Oke, 2002):

$$Q_{G,i} = R_{n,i}a_{1,i} + \left(\frac{\partial R_{n,i}}{\partial t}\right)a_{2,i} + a_{3,i} \tag{5}$$

where $\frac{\partial R_{n,i}}{\partial t} = 0.5(R_{n,i,(t-1)} - R_{n,i,(t+1)})$ and the three $a$ coefficients are defined using cited values for each surface (see Table 1). The $a$ coefficients capture the hysteresis pattern commonly observed between the $R_n$ and $Q_{G,i}$ in urban areas. See Grimmond and Oke (1999) for a full description of the OHM and the role of the $a$ parameters in $Q_{G,i}$ calculations. The $Q_{G,i}$ is then used to calculate the $T_{surf}$ for each land cover type using the force-restore method.

## 2.5 Surface temperature calculation ('force-restore')

The force-restore method is an efficient method for calculating surface temperature (Bhumralkar, 1975; Deardorff, 1978), and is an alternative to multilayer conduction approaches used in other climate models. The force-restore method is used to ensure

Table 1. Parameter setup for all TARGET simulations in this article.

| | Roof & wall‡ | Asphalt | Water | Soil (water)* | Concrete | Dry grass | Irrigated grass | Tree |
|---|---|---|---|---|---|---|---|---|
| $\alpha$ | 0.15[1] | 0.08[1] | 0.10[1] | N/A | 0.20[1] | 0.19[3] | 0.19[3] | 0.10[1] |
| $\epsilon$ | 0.90[1] | 0.95[1] | 0.97[1] | N/A | 0.94[1] | 0.98[2] | 0.98[2] | 0.98[1] |
| $C$ ($\times 10^6$) | 1.25[2] | 1.94[1] | 4.18[1] | 3.03[1] | 2.11[1] | 1.35[3] | 2.19[3] | N/A |
| $\kappa$ ($\times 10^{-6}$) | 0.05† | 0.38[1] | 0.14[1] | 0.63[1] | 0.72[1] | 0.21[3] | 0.42[3] | N/A |
| $T_m$ | 25.0 (28.2) | 26.0 (29.0) | 25.0 (24.5) | 25.0 (24.5) | 26.0 (27.9) | 20.0 (22.4) | 20.0 (21.5) | N/A |
| OHM $[a_1, a_2, a_3]$ | [0.12,0.24,-4.5][3] | [0.36,0.23,-19.3][4,5] | N/A | N/A | [0.67,0.31,-31.45][4,5] | [0.21,0.11,-16.10][6] | [0.27,0.33,-21.75][6,7] | N/A |
| s | 0.0 | 0.0 | N/A | N/A | 0.0 | 0.2 | 1.0 | N/A |
| $\beta$ | 3 | 3 | 3 | 3 | 3 | 3 | 3 | N/A |

[1] Oke (1987)

[2] Stewart et al. (2014)

[3] Järvi et al. (2014)

[4] Narita et al. (1984)

[5] Asaeda and Ca (1993)

[6] Grimmond et al. (1993)

[7] Doll et al. (1985)

$\alpha$ = surface albedo

$\epsilon$ = surface emissivity

$C$ = volumetric heat capacity (J m$^{-3}$ K$^{-1}$) ($\times 10^6$)

$\kappa$ = thermal diffusivity (m$^2$ s$^{-1}$) ($\times 10^{-6}$)

$T_m$= average soil (ground) temperature ($^\circ$C)

$\alpha_{pm}$ = LUMPS empirical parameter (alpha parameter) - relates to surface moisture

$\beta$ = LUMPS empirical parameter (beta parameter) $T_m$ bracketed values were used in Mawson Lakes suburb simulations — derived from 1 month spin-up.

*soil layer beneath water layer.

† the traditional force-restore method is not well suited to urban surfaces (e.g. roof and walls) — we use an artificially low thermal diffusivity to represent a thin layer. This is discussed further in Sect. 2.5.

‡ roof and wall layers are represented with the same model parameters.

that the model remains computationally efficient. The ground layer is conceptually divided into two layers with uniform vertical temperature: a thin surface layer and deep soil layer. The forcing term, which is driven by $Q_{G,i}$, heats the surface layer. The restore term, driven by deep soil temperature, dampens the forcing term. The change in surface temperature $T_{surf}$ for surface $i$, with respect to time $(t)$, is calculated as (Jacobs et al., 2000):

$$5 \quad \frac{\partial T_{surf,i}}{\partial t} = \frac{Q_{G,i}}{C_i D} - \frac{2\pi}{\tau}(T_{surf,i,[t-1]} - T_{m,i,[t-1]}) \quad (6)$$

where $C_i$ is the volumetric heat capacity (J m$^{-3}$ K$^{-1}$), $\tau$ is the period (86400 seconds), $D$ is the damping depth of the diurnal temperature wave $D = 2\kappa/\omega 0.5$, $\omega = 2\pi/\tau$, and $\kappa$ represents thermal diffusivity. The average soil (ground) temperature (°C) $(T_m)$ is calculated using:

$$\frac{\partial T_{m,i}}{\partial t} = \frac{\Delta Q_{G,i}}{C_i D_y} \quad (7)$$

where $D_y = D\sqrt{365}$, the damping depth for the annual temperature cycle (m).

    The force-restore method, which assumes two layers each of uniform temperature, cannot be applied to more complex surfaces such as water, trees, walls, or roofs. For roofs we set $C$ at realistic value and use $\kappa$ as a tuning parameter to represent layers of thermally active mass characteristic of most building roofs, which are often thinner than ground level surfaces. This approach produces accurate $T_{surf,roof}$ results (see Sect. 3.2), but ongoing work is needed to represent roofs in a physically

realistic and efficient manner. For simplicity, the wall surfaces are assumed to have the same thermal properties as roofs. For trees, we assume that $T_{surf,tree}$ is equal to $T_a$ (see Fig. 13 for justification), and a simple water body model is used to calculate $T_{surf,watr}$.

## 2.6   Simple water body model

The water model is used for modelling small inland water bodies, such as lakes and wetlands. Our analysis suggests that the

OHM-force-restore method cannot be used to reliably reproduce water surface temperatures. We tested the OHM modifications and parameters used by Ward et al. (2016) and found substantial over-predictions of surface water temperature (over 10 °C) during the day. As such, we developed a simple water body model to stand in for the OHM-force-restore method. The water model in TARGET is based on a single water layer, overlaying a soil layer. Essentially, the force-restore surface temperature model is implemented, and is overlain by a homogeneous mixed water layer (i.e. neglecting thermal stratification) representing

a water body of depth $d_{watr}$ (m). The model is designed to apply to water bodies of 0.1–1.0 m depths. The water model is based on the pan evaporation model of Molina Martínez et al. (2006), which closely follows that of the lake model of Jacobs et al. (1998). The water body model also determines the surface energy balance of the water surface. The energy balance model for the water layer is given by Molina Martínez et al. (2006):

$$S_{ab} + L_n + Q_{H,watr} - Q_{E,watr} - Q_{G,watr} - \Delta Q_{S,watr} = 0 \quad (8)$$

where $S_{ab}$ is absorbed shortwave radiation (W m$^{-2}$), $L_n$, the net longwave radiation (W m$^{-2}$), $Q_{G,watr}$ is the convective heat flux at the bottom of the water layer and into the soil below (W m$^{-2}$), and $\Delta Q_{S,watr}$ is the change in heat storage of the water layer (W m$^{-2}$). Solar radiation penetrates the water surface and is absorbed as described by Beer's Law (Molina Martínez et al., 2006):

$$S_{ab} = K_n\left[\beta_k + (1 - \beta_k))(1 - e^{-\eta})\right] \tag{9}$$

where $K_n$ is the net shortwave radiation (W m$^{-2}$), $\beta_k$ is the amount of shortwave radiation immediately absorbed by the water layer (set to 0.45) (Molina Martínez et al., 2006), and $\eta$ the extinction coefficient. Here, $\eta$ is given the following from Subin et al. (2012), for the water layer with depth $d_{watr}$ $(m)$:

$$\eta = 1.1925d_{watr}^{-0.424} \tag{10}$$

A correction factor for the solar path length zenith angle is often applied to Eq. (9) (Molina Martínez et al., 2006) but this has been omitted from TARGET to reduce complexity.

The $Q_{G,watr}$ into the soil at the base of the water layer is given by Molina Martínez et al. (2006):

$$Q_{G,watr} = -C_{watr}\kappa_{watr}\frac{\Delta T}{\Delta d_{watr}} \tag{11}$$

where $C_{watr}$ is the volumetric heat capacity of water (4.18×10$^6$ J m$^{-3}$ K$^{-1}$), $\kappa_{watr}$ is the eddy diffusivity of water (m$^2$ s$^{-1}$), and the change in depth $\Delta d_{watr} = d_{watr}$ (the depth of the water layer). $\kappa_{watr}$ is a complex function accounting for thermal stratification of water and surface friction velocity. To reduce complexity and assuming a mixed homogeneous water layer, a constant $\kappa_{watr}$ has been selected based on shallow lakes reported in Salas De León et al. (2016). The change in temperature $\Delta T$ (°C) is the difference between the water temperature $T_{surf\,watr}$ (°C) and the soil temperature beneath the water layer $T_{soil}$ (°C). $T_{soil}$ is calculated using the force-restore model where $Q_{G,watr}$ is equivalent to $Q_{G,i}$ in Eq. (6):

$$\frac{dT_{soil}}{dt} = \frac{(Q_{G,watr} + (K_n - S_{ab}))}{C_{watr}D} - \frac{2\pi}{\tau}(T_{soil,[t-1]} - T_{m,[t-1]}) \tag{12}$$

To represent the radiation that is not absorbed by the water, but is absorbed by the underlying soil layer, $K_n - S_{ab}$ is added to $Q_{G,watr}$.

The latent heat flux ($Q_{E,watr}$) (W m$^{-2}$) is given by Arya (2001):

$$Q_{E,watr} = \rho v L_v h_v U_z (q_s - q_a) \tag{13}$$

where $\rho v$ is the density of moist air (kg m$^{-3}$), $L_v$ is the latent heat of vaporisation (=2.43 MJ kg$^{-1}$), $h_v$ is bulk transfer coefficient for moisture $(1.4 \times 10^{-3})$ (Hicks, 1972; Jones et al., 2005), $U_z$ is the reference wind speed, $q_s$ the saturated specific humidity at $T_{surf\,watr}$, and $q_a$ is the specific humidity of the air for the given $T_a$.

The sensible heat flux above the water surface is given by Molina Martínez et al. (2006):

$$Q_{H,watr} = \rho_a C_p h_c U_z (T_a - T_{surf,watr}) \tag{14}$$

where $\rho_a$ is the density of dry air (=1.2 kg m$^{-3}$), $C_p$ the specific heat of air (1013 J kg$^{-1}$ K$^{-1}$), and $h_c$ the bulk transfer coefficient for heat ($h_c = h_v$).

Returning to Eq. (9), net long wave radiation $L_n = R_n - K_n$, leaving $\Delta Q_{S,watr}$ from the energy balance equation, which is defined as (Molina Martínez et al., 2006):

$$\Delta Q_{S,watr} = C_{watr} d_{watr} \frac{\Delta T_{surf,watr}}{\Delta t} \tag{15}$$

where $\Delta t$ is change in time (seconds) and $C_{watr}$ is the volumetric heat capacity of water (J m$^{-2}$ K$^{-1}$). Solving for $\Delta T_{surf,watr}$ and adding the change in temperature to the previous time step ($T_{surf,watr[t+1]} = T_{surf,watr,[t]} + \Delta T_{surf,watr}$) gives the new water layer temperature.

## 2.7 Calculation of urban canopy layer air temperature ($T_{ac}$)

To calculate $T_{ac}$ we first calculate a domain $T_b$ for each timestep. Assuming air temperature at 3 times building height (3H) is consistent between the neighbourhood of interest and the reference weather station location, we extrapolate reference air temperature at measurement height to 3H assuming a constant flux layer and using a bulk Richardson number-based approximation (Mascart et al., 1995). Through this simple calculation we define a domain constant $T_b$ with basic representation of atmospheric stability in TARGET.

The canyon air temperature is then calculated using a modified version of the canopy air temperature equation from the Community Land Model Urban (CLMU) (Oleson et al., 2010):

$$T_{ac} = \frac{\sum_i^7 (T_{surf,i} c_s F_i) + \left[ \frac{T_{surf,roof}}{(\frac{1}{c_s} + \frac{1}{c_a})} F_{roof} \right] + (T_b c_a W)}{\sum_i^7 (c_s F_i) + \left[ \frac{F_{roof}}{(\frac{1}{c_s} + \frac{1}{c_a})} \right] + (c_a W)} \tag{16}$$

where $F_i$ and $T_{surf,i}$ are the 2-D fractional coverage and surface temperature of surface $i$ in the canyon, $c_s$ is the conductance from surface to urban canopy layer (m s$^{-1}$), and $c_a$ is the conductance from urban canopy to the above canopy surface layer (m s$^{-1}$). In Eq. 16 we assume roofs are connected to the canyon via two resistances in series, thus representing the additional impediment to transfer of heat from a rooftop into the canyon. We hypothesized that the heat transfer from roofs to the canyon air could be approximated by two resistances in series (the canyon-to-atmosphere resistance ($c_a$) and surface to canyon resistance ($c_s$)). The logic here is that resistance to heat transfer from the roof surface to the canyon should be greater than $c_a$

or $c_s$ independently. Through sensitivity testing we were able to demonstrate that this assumption improves predicted canyon air temperature. The $c_a$ is calculated following Masson (2000) and using the stability coefficients from Mascart et al. (1995). The $c_s$ term is from Masson (2000):

$$r_s = \frac{\rho_a C_p}{11.8 + 4.2 U_{can}} \tag{17}$$

5    where $c_s = \frac{1}{r_s}$ and $U_{can}$ is the wind speed in canyon (m s$^{-1}$) (Kusaka et al., 2001):

$$U_{can} = U_{top} exp\left(-0.386\frac{H}{W}\right) \tag{18}$$

where $U_{top}$ is the wind speed at the top of the canyon (m s$^{-1}$). $U_{top}$ is estimated at 3H based on the observed wind speed at a nearby observational site (ideally an airport) using a logarithmic relationship. Airports are relatively devoid of roughness elements and wind speed is typically measured at 10 m above the surface. As such, the assumption a logarithmic profile through 10   the roughness sublayer (Masson, 2000) is imposed.

## 3   Methods and data

### 3.1   Overview

As part of the model evaluation, we conducted a range of simulations that test model performance for both $T_{surf}$ and $T_{ac}$. These validation experiments are focused on clear sky summertime conditions. Clear sky was chosen because the local-cooling 15   effects of GBI are most notable during these conditions. First, we tested the model's ability to simulate $T_{surf}$ for each land cover type that can be prescribed in TARGET (i.e. dry grass, asphalt etc.), using ground-based observations of $T_{surf}$ (Sect. 3.2). These simulations by land cover type, provide a detailed assessment of model parameters, and the underlying energy balance dynamics and resulting $T_{surf}$ for each land cover class. Second, we conducted suburb scale simulations of Mawson Lakes, Adelaide, for which we have high resolution remotely sensed $T_{surf}$ observations and in situ $T_{ac}$ data (Sect. 3.3). The 20   suburb scale simulations reflect the way the model is intended to be used by practitioners.

### 3.2   Land cover simulations

To test model performance at simulating $T_{surf}$ of different land cover classes and perform sensitivity analysis on a number of model parameters, we used ground-based observations of $T_{surf}$ from the Melbourne metropolitan area. Coutts et al. (2016) deployed infrared temperature sensors (SI-121 - Apogee), during February 2012 (5 min averages), across a range of land 25   cover types including: asphalt, concrete, grass, irrigated grass, steel roof, and water. Infrared sensors were mounted above the aforementioned surface types installed at heights of approximately 1.5–2 m. The conditions during this period represented near-typical summertime conditions in Melbourne; including a number of days (15th, 24th, and 25th February) where air

temperature exceeded 30°C (see Fig. 11). These hotter days were characterised by northerly winds, which bring hot and dry air from Australia's interior, and often result in heatwave conditions in Melbourne. Additionally, there was at least one cloudy day where incoming shortwave radiation ($K \downarrow$) dropped significantly and negligible amount of rainfall occurred (17th February) To compare the Coutts et al. (2016) observations with TARGET we ran the model for each surface type (i.e. 100 % grass or roof etc) with radiation forcing data from the Melbourne Airport weather station during the time period in question. The $T_b$ calculation was not needed since we only calculated $T_{surf}$ for this part of the model evaluation. The 30 min output from TARGET was compared with $T_{surf}$ observations and statistics were calculated.

## 3.3 Suburb scale simulations (Mawson Lakes)

In addition to the land cover category testing, we also conducted suburb scale simulations of $T_{surf}$ and $T_{ac}$ for Mawson Lakes, Adelaide (Fig. 3). The suburb scale simulations used observational data from the Mawson Lakes field campaign, conducted 13-18th February 2011, which represented average summertime conditions in Adelaide (Broadbent et al., 2018b). For these simulations, the model was run on 30 m ($T_{surf}$) and 100 m ($T_{ac}$) grids over the Mawson Lakes suburb for the period 13th–18th February (Fig. 12). Remotely sensed land cover data from the campaign were used to define land cover, and building morphology was defined using LiDAR data (see Broadbent et al. (2018b)). The Mawson Lakes simulations used the same parameter setup as above (summarised in Table 1), and were forced with meteorological data from the Kent Town Bureau of Meteorology (ID 023090) weather station. Modelled $T_{surf}$ was validated using observed remotely sensed $T_{surf}$ (night - 15th February and day - 16th February), which was resampled to 30 m resolution (Broadbent et al., 2018b). To validate $T_{ac}$, we use data from 27 automatic weather station (AWS) that were also deployed during the Mawson Lakes field campaign (see Fig. 3 for AWS locations).

## 4  Model evaluation results and discussion

### 4.1  Land cover simulations

The surface temperature for each land cover class was simulated for a 14 day period during February 2012. The results show that modelled surface temperature for all 3 impervious surfaces (concrete, asphalt, and roof) were reasonably well predicted with mean bias errors (MBE) of 0.88, -0.22, and -1.16°C, respectively (Fig. 4a–f). The root mean square error (RMSE) values for impervious surfaces were around 3.5–4°C. These RMSE values represent about 15% of diurnal $T_{surf}$ variation, which implies good model skill given the simplicity of the approach.

The night of the 16 February was not well captured at the concrete and asphalt sites. The $T_{surf,conc}$ and $T_{surf,asph}$ were under-predicted (up to 5°C cooler than observations) on the night of 16 February, which may have been caused by warm air advection. The TARGET approach cannot account for the effects of warm air advection on surface temperature, as there is not feedback between $T_{ac}$ and $T_{surf}$. Despite this limitation, the broad timing and magnitude of heating and cooling was well captured for all three impervious land cover types.

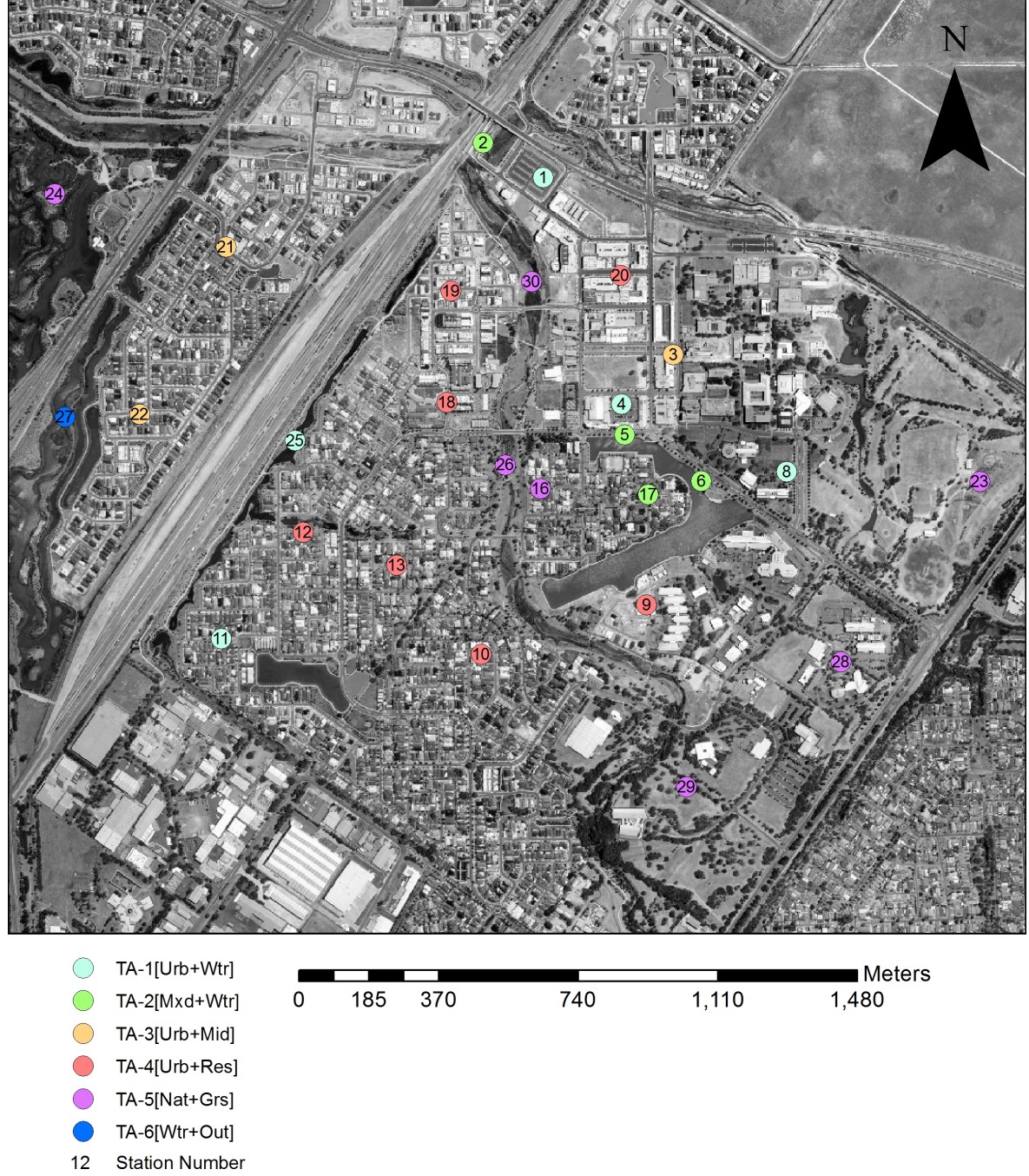

**Figure 3.** Mawson Lakes suburb with weather station locations. The numbers indicate individual weather stations while the color coding specifies groups of sites with statistically similar thermal characteristics. The names of each cluster indicate the average land surface characteristics: urban sites with nearby water (TA-1[Urb+Wtr]), mixed land-use with nearby water (TA-2[Mxd+Wtr]), urban mid-rise type sites (TA-3[Urb+Mid]), urban residential sites (TA-4[Urb+Res]), natural grass dominated sites (TA-5[Nat+Grs]), and a single outlier site (TA-6[outlier]) (Broadbent et al., 2018b).

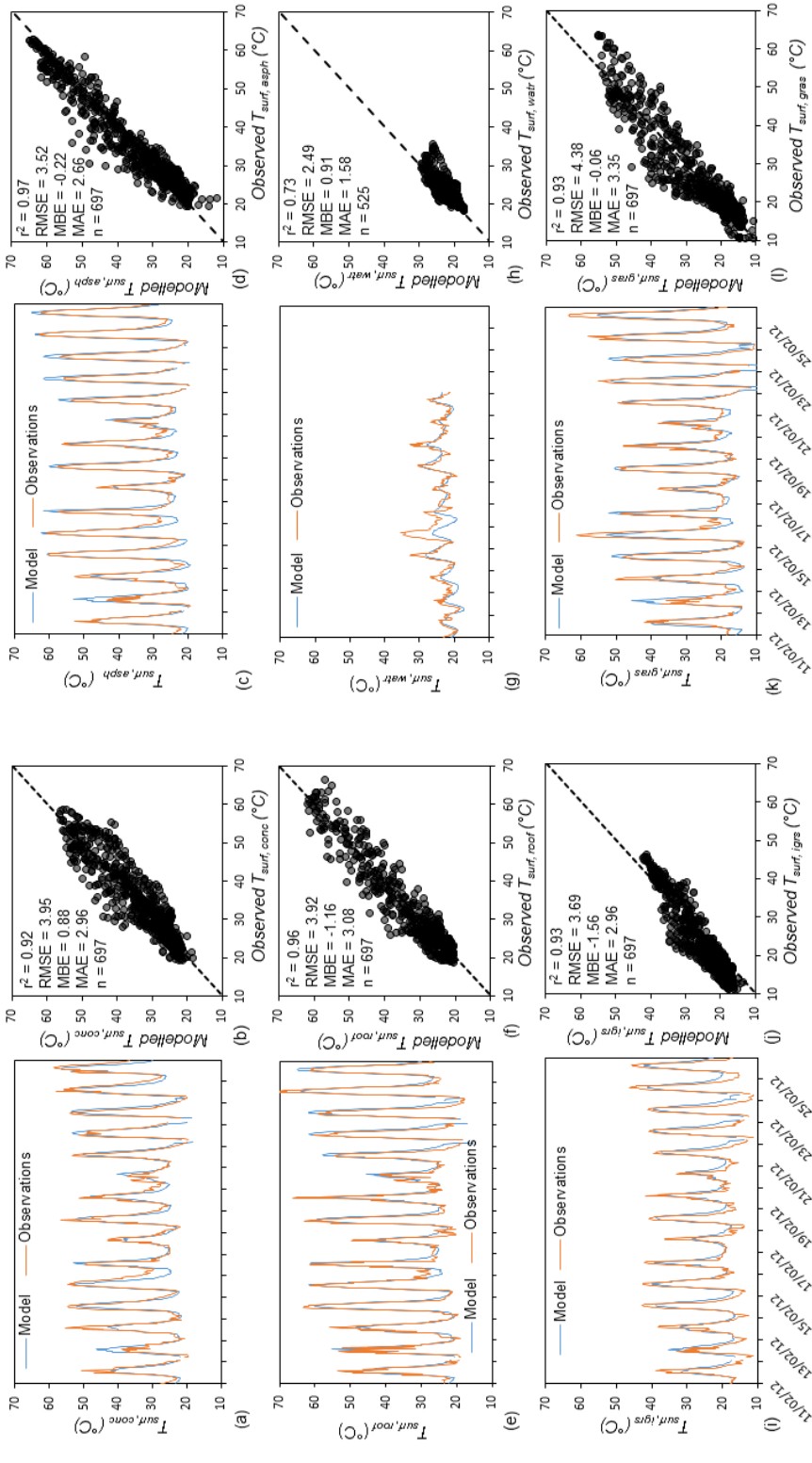

**Figure 4.** Observed *vs.* modelled (a/b) $T_{surf,conc}$, (c/d) $T_{surf,asph}$, (e/f) $T_{surf,roof}$, (g/h) $T_{surf,watr}$, (i/j) $T_{surf,irgs}$, and (k/l) $T_{surf,gras}$. All time series plots are for the period 11 – 25 February. Note that the water site only had observational data for the period 11 – 21 February due to instrument failure. $r^2$ = correlation coefficient, RMSE = root mean square error, MBE = mean bias error, and MAE = mean absolute error.

Model performance for $T_{surf,watr}$ had a low MBE of 0.91°C but the $r^2$ value of 0.76 suggests the model captured diurnal $T_{surf,watr}$ variation less accurately than other surfaces (Fig. 4g–h). In particular, daily maximum $T_{surf,watr}$ were under-predicted on hotter days (e.g. 14th February). TARGET uses a different module for water bodies (see Sect. 2.6). This simple module treats water as a single layer overlying soil. Despite the under-prediction on 14 February, the simple water body model can reproduce $T_{surf,watr}$ to an acceptable standard.

Modelled $T_{surf,irgs}$ had a MBE (-1.56°C) comparable to that of impervious surfaces (Fig. 4i–j). However, the RMSE for irrigated grass (3.69°C) represents approximately 20% of diurnal $T_{surf,igrs}$ variation, suggesting model error is slightly higher than for the impervious surfaces. Generally $T_{surf,igrs}$ was slightly over-predicted at night and under-predicted at the daily maxima. This skewing of the scatter plot suggests that thermal inertia is too high in the model. Overall, the model was skillful enough to capture the timing and amplitude of $T_{surf,igrs}$.

Dry grass had a small MBE error (0.06°C), but the largest RMSE of the surfaces tested (4.38°C). However, this RMSE only equated to approximately 10–15% of diurnal $T_{surf,gras}$ variability (Fig. 4k–l) as dry grass had the largest amplitude of surface temperature variability. Dry grass exhibited the same skewing in the scatterplot as irrigated grass, with a general over-prediction of night-time temperatures and under-prediction of daytime maxima.

## 4.2 Suburb scale simulations (Mawson Lakes)

**Surface temperature**

In addition to the land cover simulations we conducted suburb scale modelling of the Mawson Lakes site. These simulations reveal how the TARGET model can be operationalised by practitioners who want to assess the cooling benefits of blue infrastructure or greening initiatives. Suburb scale simulations were conducted using the same parameters as above (Table 1). We run the model at 30 m spatial resolution for $T_{surf}$ simulations and 100 m for simulations of $T_{ac}$. Fig. 5 shows the predicted $T_{surf}$ for the Mawson Lakes domain plotted against observed $T_{surf}$. The Mawson Lakes simulations revealed the initial conditions of the $T_m$ parameter (which represents the average temperature in the ground layer) are important for good model performance. A spin-up period (1 month) had to be used to obtain initial $T_m$ values for each surface type. This can be quickly and easily achieved by running the force-restore module for a single point for each surface type. A future version of the model will automatically spin-up initial $T_m$ values. The model output also shows that some of input land cover is poorly categorised, resulting in population of grid points where modelled $T_{surf}$ is over-predicted. Additionally, errors in the observed $T_{surf}$ caused by heterogeneity of roof emissivity, also contribute to apparent inaccuracies of modelled $T_{surf}$. In general, the daytime $T_{surf}$ was slightly over-predicted and the complexity of spatial variability was not fully captured. However, this is a positive result given only 8 land cover classes are represented in the model. Overall, the daytime $T_{surf}$ was well predicted with the range and magnitude of $T_{surf}$ captured by the model.

The results suggest that night-time $T_{surf}$ was under-predicted by model. The range of modelled nocturnal $T_{surf}$ variability (c. 8°C) was much smaller than observed variability (c. 18°C). This under-prediction of variability could reflect the fact that some processes that dictate the rate of nocturnal cooling are not fully accounted for in this approach. Nevertheless, the general

spatial patterns of $T_{surf}$ are captured well. Further, given that the range of $T_{surf}$ is smaller at night, this under-prediction is of minimal consequence for modelled $T_{ac}$. The nocturnal $T_{surf}$ of impervious surfaces was also under-predicted in the land cover simulations (i.e. Sect. 4.1) under warm advection conditions. Although warm advection conditions were not observed during the Mawson Lakes campaign, it is worthwhile further investigating this phenomenon, in future work, to negate its effect
and improve nocturnal $T_{surf}$ accuracy.

**Air temperature**

Spatial plots of modelled 3 am and 3 pm $T_{ac}$ are shown in Fig. 6. The modelled air temperatures are biased towards warmer air temperature in urban areas and cooler air temperature in rural areas. These biases are partly driven by the lack of advection in the model. Without atmospheric mixing, the local impacts of pervious and impervious surfaces are exaggerated causing an
additional cooling and warming effect in rural and urban areas, respectively However, the general patterns of $T_{ac}$ are reasonable and as expected. We also extracted modelled $T_{ac}$ from the grid points where the 27 AWS were located (grid points were centred at the AWS) for a 2 day period (15-16th February 2011) (Fig. 7). The $T_{ac}$ was generally well predicted (Fig. 7), with a RMSE of 2.0 °C. These results are about the same accuracy as simulations, from the same site, conducted using a more sophisticated and computationally expensive urban climate model called SURFEX (Broadbent et al., 2018a). Although a simple model,
TARGET appears as accurate as more complex models. Additionally, TARGET does not require the user to provide above canyon forcing data (e.g. $T_b$), which is needed for other models and is not easily obtained. TARGET tended to over-predict average $T_{ac}$ at all urban sites (Fig. 8). Residential sites (TA-4[Urb+Res] cluster [red]) were too warm during the day. This over-prediction is likely due to the uniform wall and roof thermal parameters used, which are not representative of residential areas. Further, the lack of horizontal mixing may have exacerbated warmer temperatures in these areas. By contrast, the TA-5[Nat+Grs]
cluster is too cool at night. The model predicts the formation of a stable layer with cool air trapped near the surface. Overall, the diurnal range and average $T_{ac}$ are well captured by the model.

    Finally, there is some hysteresis in Fig. 6, indicating that modelled $T_{ac}$ is slightly out-of-sync with observed $T_{ac}$. This could be due to the approach used to diagnose $T_b$, which assumes a constant $Ri$ in the surface layer, and therefore heats up too quickly during the morning. Improvement in the $T_b$ term is an area for future model development. However, we believe it is important
that TARGET calculates $T_b$, as this makes the model much more accessible to non-expert users. Given the simplicity and computational efficiency of the model approaches used, TARGET shows good skill for predicting urban $T_{ac}$. Overall, the air temperature evaluation shows we can have confidence in the accuracy of the model and its potential to be used by practitioners.

## 5   Heat mitigation scenarios

To demonstrate how TARGET can be used by practitioners to predict GBI cooling impacts, two simple heat mitigation scenarios
are presented: (1) a doubling of existing tree cover ('2×TREE') (Fig. 9) and (2) all dry grass is converted to irrigated grass ('IRRIGATION') (Fig. 10). The '2×TREE' scenario assumes a maximum tree coverage of 75%. The results presented here

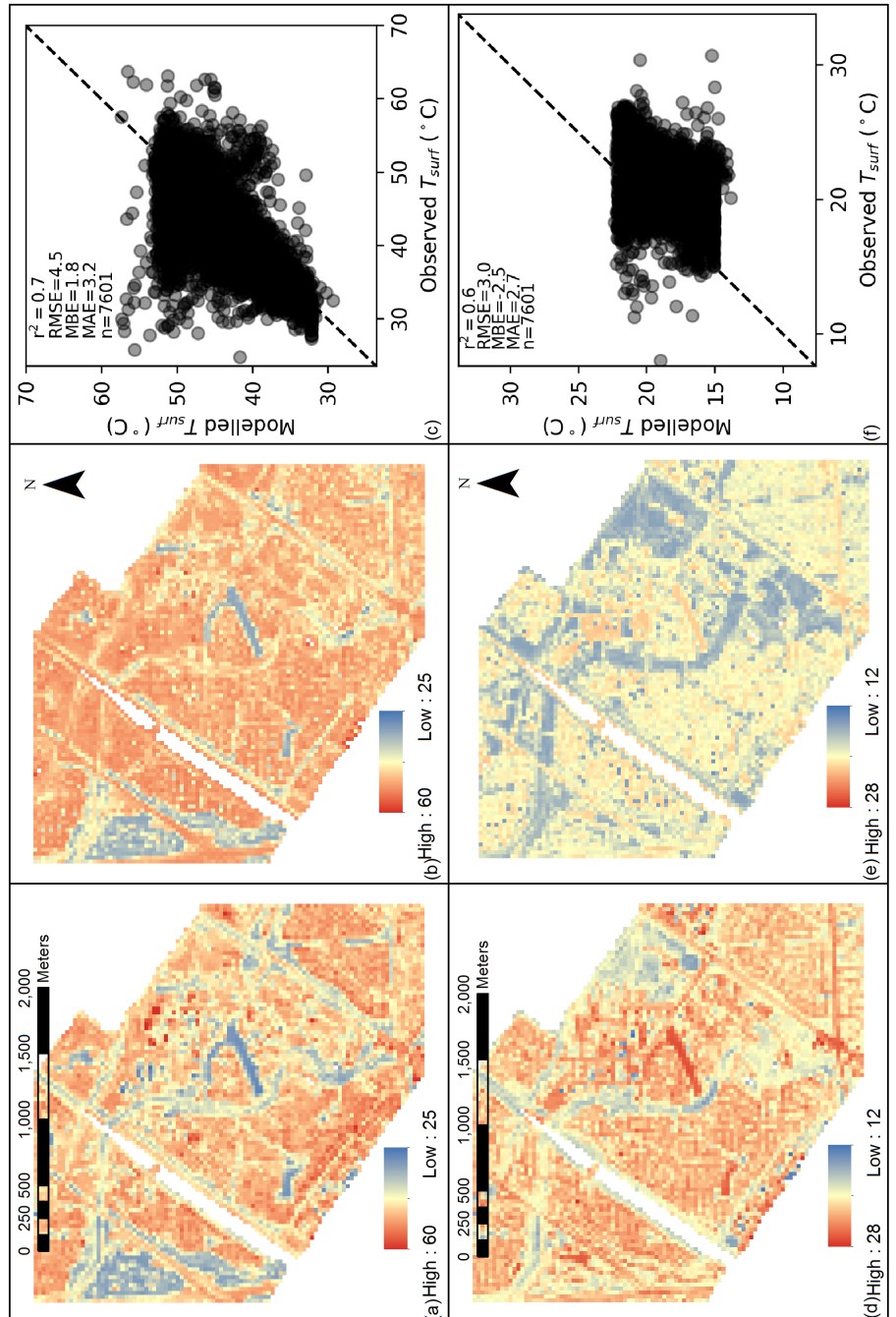

**Figure 5.** Mawson Lakes observed (a & d) $T_{surf}$ and modelled (b & e) $T_{surf}$ for day (TOP) and night (BOTTOM). Areas where land cover is categorized as 'other' were not simulated. Note: $T_{surf}$ here does not include $T_{surf,wall}$ for comparison with horizontally averaged aerial imagery observations.

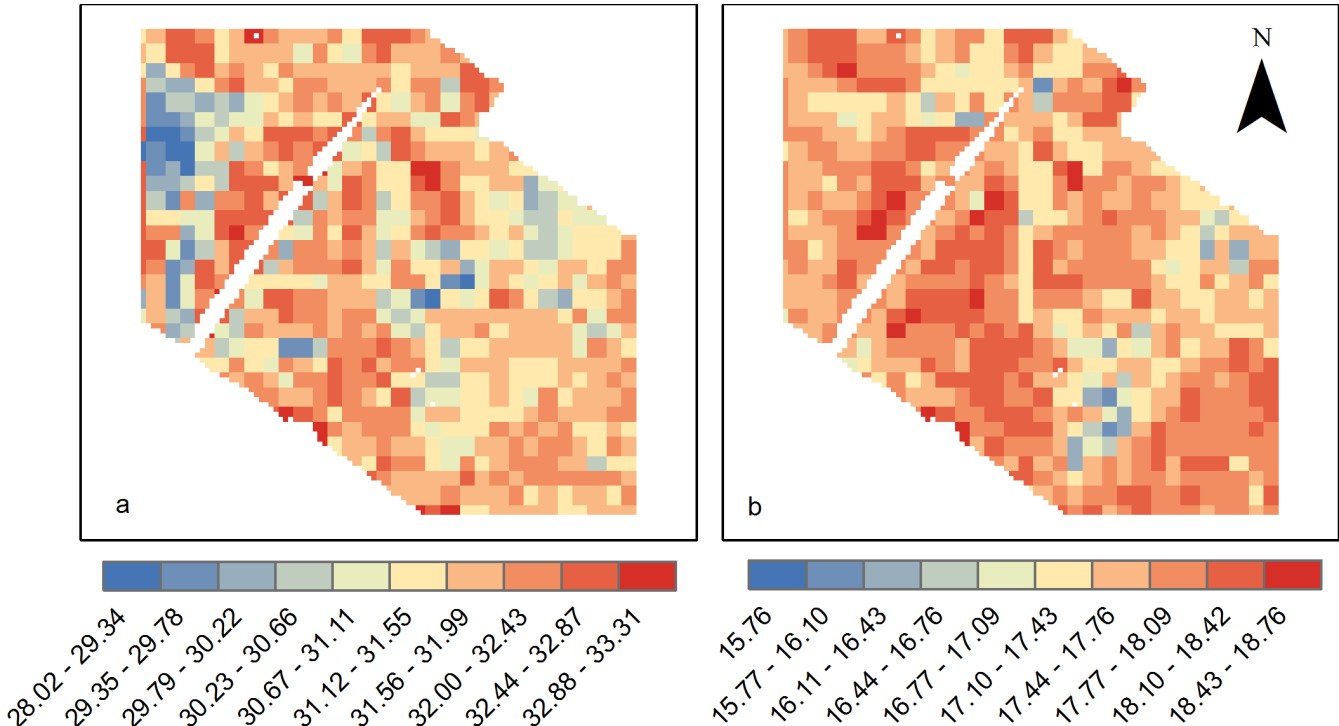

**Figure 6.** Spatial map of modelled $T_{ac}$ (30 m) for (a) day (3 pm) and (b) night (3 am) in Mawson Lakes domain. Points with $F_{roof} > 0.75$ are excluded.

represent the local maximum cooling potential of GBI. In reality, the cooling local magnitude will be decreased by advection, which TARGET does not represent.

The 2×TREE scenario shows maximum cooling of 3.0 °C during the day and smaller effect ($< 0.25°$C) at night (Fig. 9). The IRRIGATION scenarios suggests that increasing irrigation can have a small warming effect ($< 0.75°$C) at night, and cooling of up to 1.75°C at 3 pm (Fig. 10). The amount of land cover change differs in each scenario. As such, we calculate the cooling sensitivity ($\gamma$) as:

$$\gamma = \left(\frac{\Delta T_{ac}}{\Delta LC}\right) \times 0.10 \tag{19}$$

where $\Delta LC$ is the average land cover change (fraction) (Table 2); this metric demonstrates the average $\Delta T_{ac}$ per 10% surface change. Model results suggests that trees are about 2.5 times more effective at providing cooling at 3 pm (Table 2). The results for both heat mitigation simulations are within the expected magnitudes based on previous heat mitigation modelling studies (Grossman-Clarke et al., 2010; Middel et al., 2015; Daniel et al., 2016; Broadbent et al., 2018a). These simulations demonstrate that TARGET not only reproduces observations accurately, but can be used with confidence to efficiently assess the efficacy of heat mitigation measures.

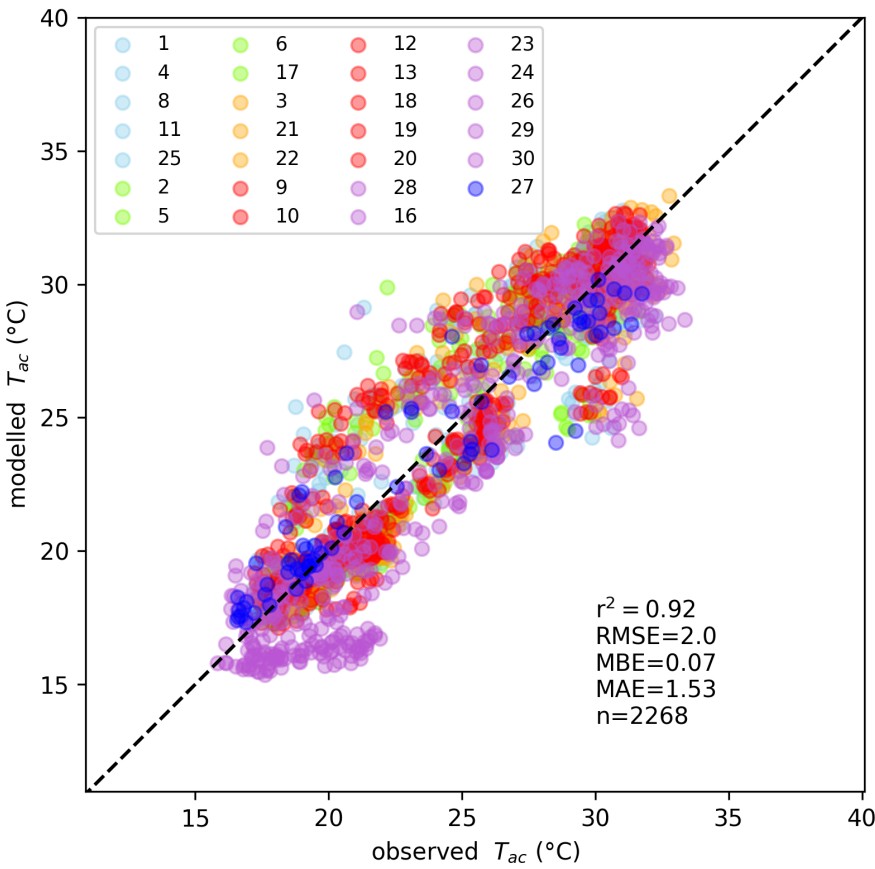

**Figure 7.** Modelled $T_{ac}$ vs observed $T_{ac}$ for Mawson Lakes weather stations (15–16th February 2011, 30 minute data). The numbers and colors correspond to individual stations and clusters shown in Fig. 3.

**Table 2.** Summary of domain average cooling impacts (°C) for BGI heat mitigation scenarios.

| Scenario | $\Delta T_{ac}$ 3 am | $\gamma$ 3 am | $\Delta T_{ac}$ 3 pm | $\gamma$ 3 pm | $\Delta T_{ac}$ daily | $\gamma$ daily |
|---|---|---|---|---|---|---|
| 2×TREE | -0.13 | -0.10 | -0.50 | -0.50 | -0.28 | -0.09 |
| IRRIGATION | 0.34 | 0.09 | -0.58 | -0.20 | -0.20 | -0.04 |

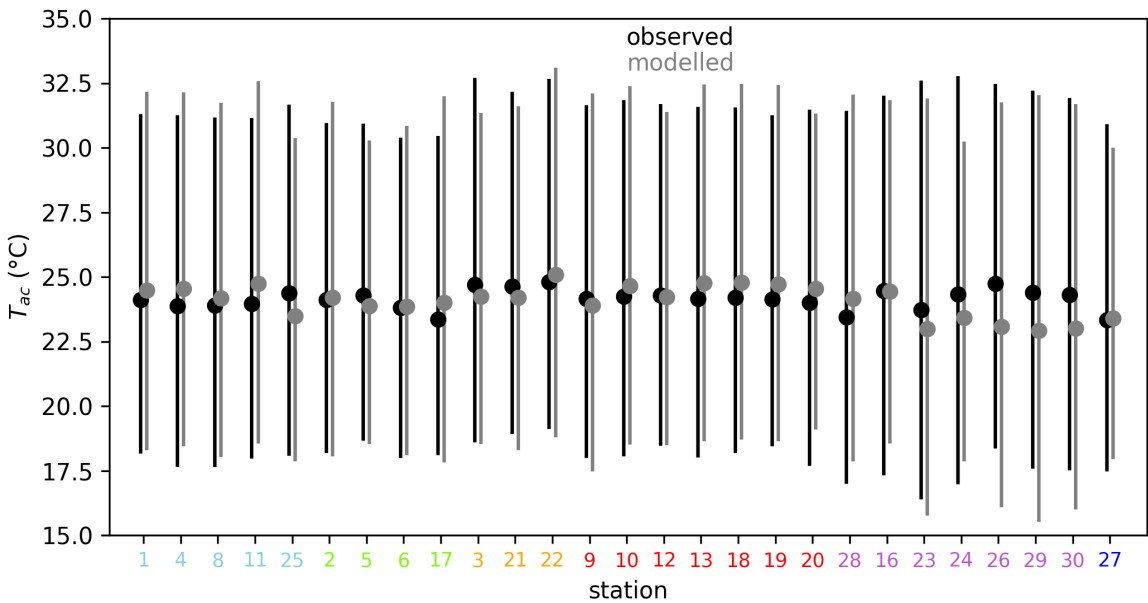

**Figure 8.** Boxplot of modelled $T_{ac}$ (grey) vs observed $T_{ac}$ (black) for Mawson Lakes with average, min, and max $T_{ac}$ shown. Boxplots were generated from 30 minute data from period 15–16th February 2011. The numbers and colors (x-axis) correspond to individual stations and clusters in Fig. 3.

## 6   Limitations of the model

As discussed above, TARGET aims to be a simple and accessible urban climate model that provides scientifically defensible and accurate urban temperature predictions. To achieve simplicity the model necessarily makes some assumptions and omissions that users should be aware of. TARGET is primarily intended to model urban temperatures during clear sky conditions. The model does not simulate rainfall and therefore should not be used for periods containing significant precipitation. Further, the model can be used to simulate street-level air temperature and surface temperature for days to weeks (i.e. a heatwave), but has not been tested or validated for longer scale simulations (i.e. months to years).

For computational efficiency, the model assumes no horizontal advection inside or above the UCL. In general, advection reduces the local impacts (i.e. cooling directly adjacent the cooling intervention) of GBI due to atmospheric mixing, and therefore we expect TARGET to provide estimates of near maximum cooling benefits. In reality, cooling effects will be diminished by advection, especially during the day and during high wind conditions.

As mentioned, the force-restore method is used for roof and wall surfaces with an artificially reduced $\kappa$ value. Although this approach generally performed well, it is our intention to develop and integrate a more realistic formula for modelling roof and wall $Q_{G,i}$. A conduction model, although more computationally expensive, would allow more flexibility as different types of

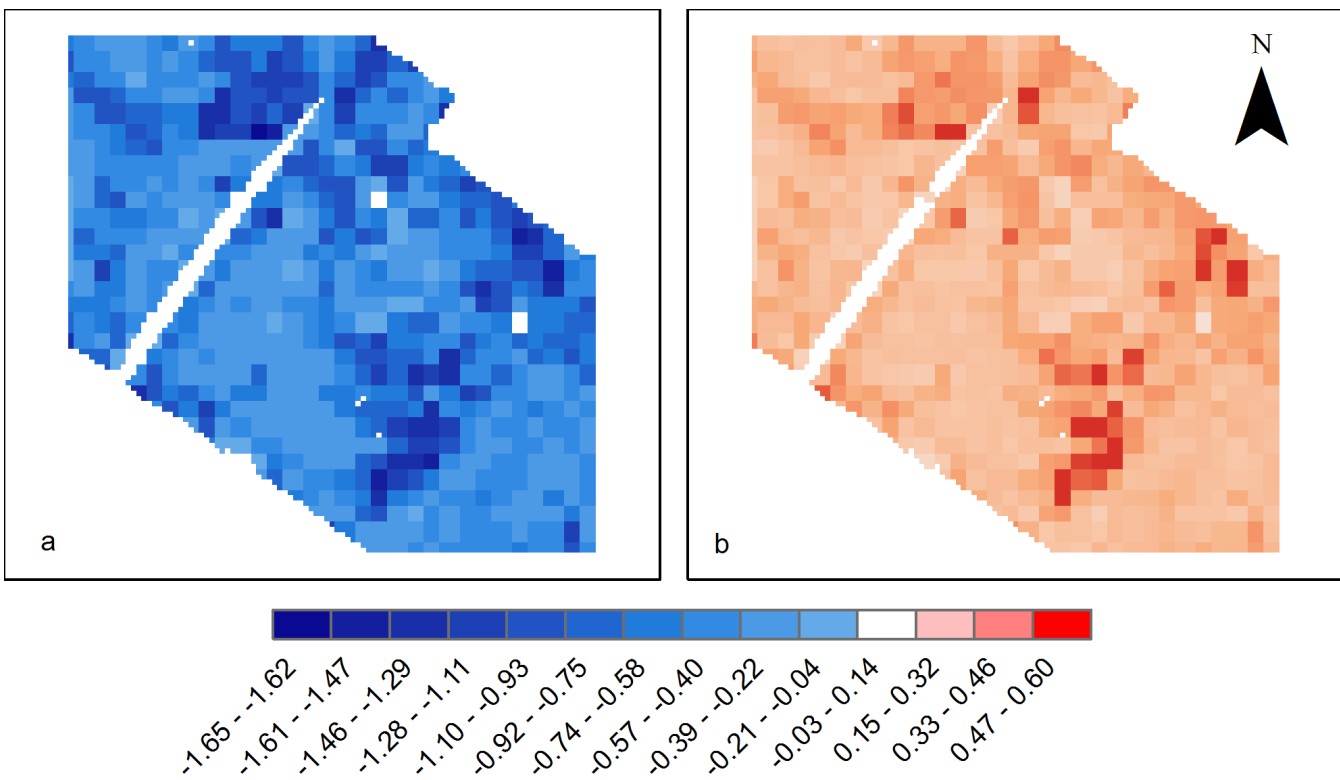

**Figure 9.** The $\Delta T_{ac}$ ($^{\circ}$C) for IRRIGATION – BASE at (a) 3 pm and (b) 3 am for Mawson Lakes domain.

roof (which do vary significantly) could be represented. Furthermore, wall surfaces are treated the same as roofs in TARGET, which is unrealistic. Improved representation of walls and roofs are key areas for future model development.

In addition, the $Q_{G,i}$ (hence the heat transfer between the urban canopy atmosphere, as the residual) is parametrized according to $R_n$ and the building parameters. This means that the dependency on the other atmospheric conditions, such as air temperature, wind speed, and humidity, is neglected in TARGET. However, given that the OHM (used to calculate $Q_{G,i}$) was developed based on observational data collected during summertime clear sky conditions, we are confident that TARGET will provide reasonable results during summer. Ongoing testing is needed to ascertain the limitations of the use of the OHM in TARGET.

The $Q_{G,i}$ calculation for water sources used a different method to other surfaces (see Section 2.6). Further, a resistance formulation is used to calculate the $Q_{H,i}$ over water bodies (see Eq. (14)), whereas $Q_{H,i}$ for the non-water surfaces is calculated as a residual (and not temperature and wind-speed dependent). These different model formulations for water may lead to artificial non-physical discrepancies. However, testing has not revealed any unexpected behavior. As TARGET is a climate-service-oriented tool, we think that good model performance is more important than the consistency of physics schemes used.

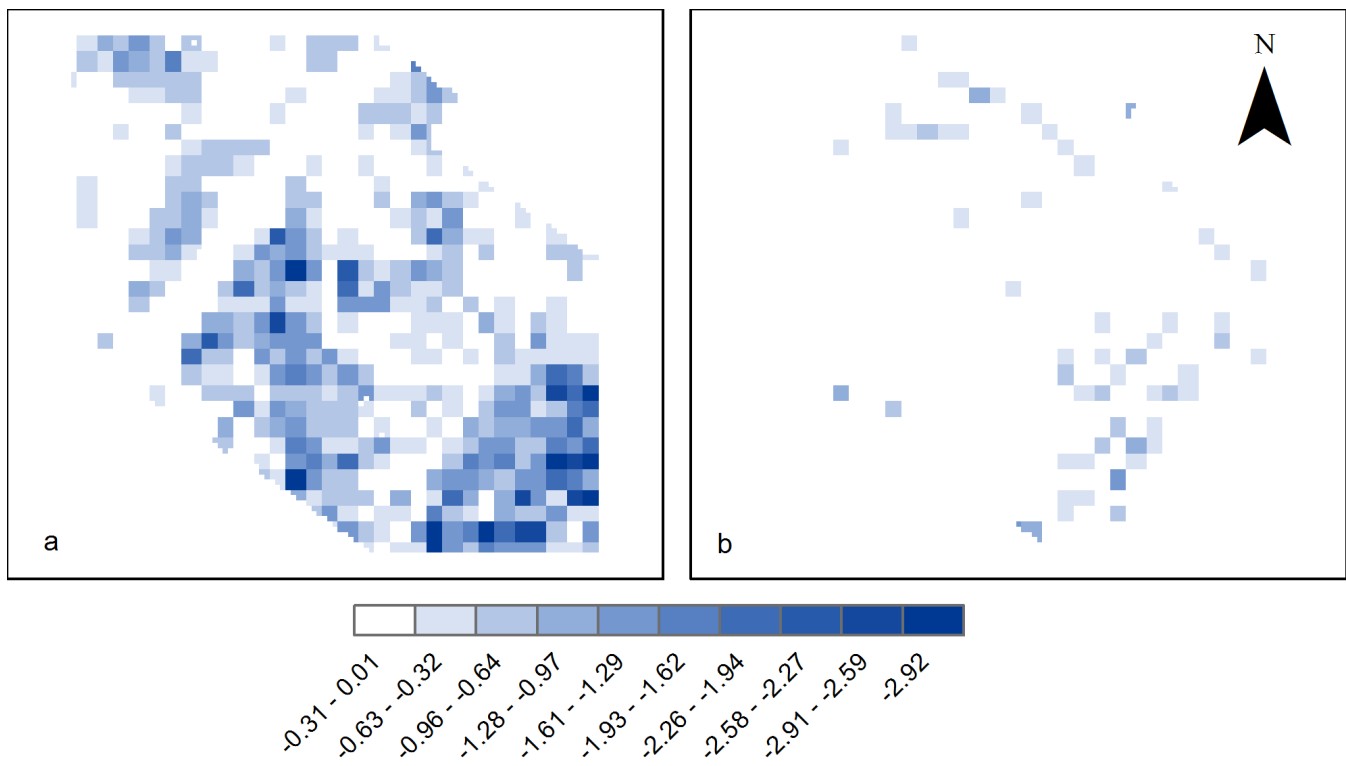

**Figure 10.** The $\Delta T_{ac}$ ($^{\circ}$C) for 2$\times$TREE – BASE at (a) 3 pm and (b) 3 am for Mawson Lakes domain.

## 7  Conclusions and future work

This paper has presented TARGET; a simple and user-friendly urban climate model that is designed to be accessible to urban planners and policy makers. The model contains a number of key limitations that are outlined above. However, despite these caveats, rigorous testing suggests TARGET shows excellent potential for modelling the cooling effects of GBI projects. We believe this novel model is well balanced between complexity and accuracy. The computational efficiency of the model and the reduced amount of input data required ensures that non-skilled users could use the model to ascertain reliable urban cooling estimates. Ongoing work will be done to improve TARGET, including the creation of GUI, the addition of human thermal comfort indices, and the improvements to model physics outlined above.

## 8  Code availability and licensing

TARGET is distributed under the Creative Commons Attribution-NonCommercial-ShareAlike 4.0 Generic (CC BY-NC-SA 4.0). TARGET code cannot be used for commercial purposes. It is available in two versions, Python or Java. The Python code can be downloaded from https://doi.org/10.5281/zenodo.1300023 or Java code is available here https://zenodo.org/record/1310138. We recommend using the Java version as it runs faster than the Python code.

# 9 List of Symbols

$a_1$ - Objective hysteresis model (OHM) parameter

$a_2$ - Objective hysteresis model (OHM) parameter

$a_3$ - Objective hysteresis model (OHM) parameter

$A_{asph}$ - land cover asphalt plan area ($m^2$)

$A_{conc}$ - land cover concrete plan area ($m^2$)

$A_{gras}$ - land cover grass plan area ($m^2$)

$A_{igrs}$ - land cover irrigated grass plan area ($m^2$)

$A_{tree}$ - land cover tree plan area ($m^2$)

$A_{roof}$ - land cover building plan area ($m^2$)

$A_{wall}$ - land cover wall plan area ($m^2$)

$A_{watr}$ - land cover water plan area ($m^2$)

$C_{watr}$ - volumetric heat capacity of water ($J\,m^{-2}\,K^{-1}$)

$c_a$ - conductance from urban canopy to the above canopy surface layer ($m\,s^{-1}$)

$c_s$ - conductance from surface to urban canopy layer ($m\,s^{-1}$)

$C_{watr}$ - volumetric heat capacity of water ($4.18 \times 10^6\,J\,m^{-3}\,K^{-1}$)

$C_p$ - specific heat of air ($1013\,J\,kg^{-1}\,K^{-1}$)

$d_{watr}$ - depth of water body (m)

$D_y$ - damping depth for the annual temperature cycle (m)

$\eta$ - extinction coefficient

$F_i$ - fraction of land cover type $i$ (%)

$H$ - average building height (m)

$h_c$ - bulk transfer coefficient for heat ($h_c = h_v$)

$h_v$ - bulk transfer coefficient for moisture ($=1.4 \times 10^{-3}$)

$K_n$ - net shortwave radiation ($W\,m^{-2}$)

$K\downarrow$ - incoming shortwave radiation ($W\,m^{-2}$)

$L_n$ - net longwave radiation ($W\,m^{-2}$)

$L\downarrow$ - incoming longwave radiation ($W\,m^{-2}$)

$L\uparrow$ - outgoing longwave radiation ($W\,m^{-2}$)

$L_v$ - latent heat of vaporisation ($=2.43\,MJ\,kg^{-1}$)

$Q_{E,watr}$ - latent heat flux for water surface ($W\,m^{-2}$)

$Q_{G,i}$ - storage heat flux for surface type $i$ from LUMPS ($W\,m^{-2}$)

$Q_{G,watr}$ - convective heat flux at the bottom of the water layer (and into the soil below) ($W\,m^{-2}$)

$Q_{H,i}$ - sensible heat flux for surface $i$ from LUMPS ($W\,m^{-2}$)

$Q_{H,watr}$ - sensible heat flux for water surface (W m$^{-2}$)

$r_a$ - resistance from urban canopy to the atmosphere (s m$^{-1}$)

$RH$ - relative humidity (%)

$Ri$ - Richardson number

$R_n$ - net radiation (W m$^{-2}$)

$S_{ab}$ - absorbed shortwave radiation (W m$^{-2}$)

$SVF$ - sky view factor

$r_s$ - resistance from surface to canopy (s m$^{-1}$)

$T_a$ - reference air temperature (°C)

$TARGET$ - CRC for Water Sensitive Cities microclimate toolkit model

$T_{ac}$ - street level (urban canopy layer) air temperature (°C)

$T_b$ - the air temperature above the urban canopy layer (°C)

$T_m$ - average soil (ground) temperature (°C)

$T_{high}$ - upper level temperature for Richardson number calcuation (°C)

$T_{low}$ - lower level temperature for Richardson number calcuation (°C)

$T_{soil}$ - soil temperature (°C)

$T_{surf}$ - surface temperature from the force-restore model (°C)

$U_{can}$ - wind speed in canyon (m s$^{-1}$)

$U_{top}$ - wind speed at the top of the canyon (m s$^{-1}$)

$U_z$ - reference wind speed (m s$^{-1}$)

$W$ - average street width (m)

$W*$ - average street width minus tree width (m)

$W_{tree}$ - tree width (m)

$W_{roof}$ - roof width (m)

$\alpha$ - surface albedo

$\alpha_{pm}$ - LUMPS empirical parameter (alpha parameter) - relates to surface moisture

$\beta$ - LUMPS empirical parameter (beta parameter)

$\beta_k$ - amount of shortwave radiation immediately absorbed by the water layer (set to 0.45)

$\Delta Q_{S,watr}$ - change in heat storage of the water layer (W m$^{-2}$)

$\epsilon$ - surface emissivity

$\kappa$ - thermal diffusivity (m$^2$ s$^{-1}$)

$\kappa_{watr}$ - eddy diffusivity of water (m$^2$ s$^{-1}$)

$\lambda_C$ - the plan area ground level surfaces (m$^2$)

$\rho_a$ - density of dry air (=1.2 kg m$^{-3}$)

$\rho v$ - density of moist air (kg m$^{-3}$)

$\sigma$ - Stefan-Boltzmann constant (5.67 × 10$^{-8}$ W m$^{-2}$K$^{-4}$)

**Author Contribution**

AMB, AMC, KAN, MD, HW, ESK, NJT assisted with model development and design. AMB conducted model evaluation and analysis. All authors contributed to the writing of the manuscript.

**Acknowledgements**

At Monash University, Ashley Broadbent and Kerry Nice were funded by the Cooperative Research Centre for Water Sensitive Cities, an Australian Government initiative. While at Arizona State University, Ashley Broadbent was supported by NSF Sustainability Research Network (SRN) Cooperative Agreement 1444758, NSF grant EAR-1204774, and NSF SES-1520803. Matthias Demuzere and Hendrik Wouters funded by the Cooperative Research Centre for Water Sensitive Cities. The contribution of Matthias Demuzere was funded by the Flemish regional government through a contract as a FWO (Fund for Scientific Research) post-doctoral research fellow. E Scott Krayenhoff was supported by NSF Sustainability Research Network (SRN) Cooperative Agreement 1444758 and NSF SES-1520803.

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

### Appendix A

#### A1 Meteorological conditions during validation periods

As outlined in Sect. 3, we conducted model validation experiments during two different periods. A summary of the meteorological conditions for land cover [Melbourne] (Fig. 11) and suburb scale [Mawson Lakes] (Fig. 12) simulations are provided below.

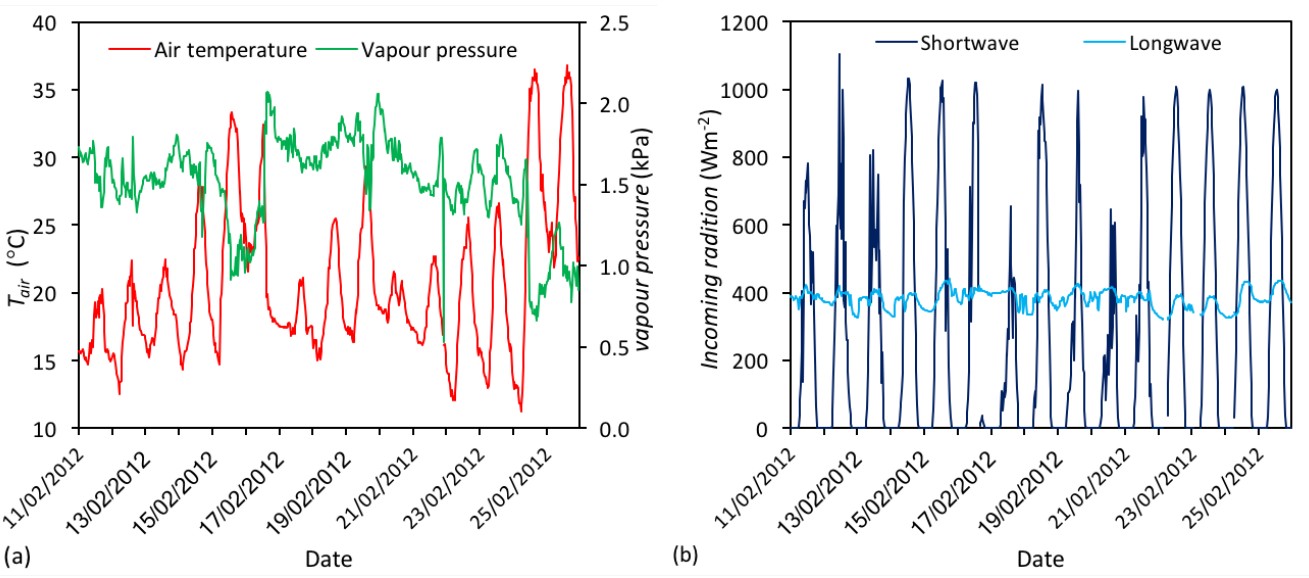

**Figure 11.** Meteorological conditions during land cover validation period. Data source: Melbourne Airport Bureau of Meteorology (ID 086282) weather station.

#### A2 Tree surface temperature

To assess $T_{tree}$ we obtained observational data from a tree experiment conducted in Melbourne, which including $T_{surf}$ observations of the tree canopy (collected during February 2014). We also obtained a BoM meteorological forcing data for the 2014 case study period. This period (not shown) was very similar to the February 2012 period (Fig. 11) used above. The tree data confirms that $T_a$ is excellent predictor or $T_{tree}$.

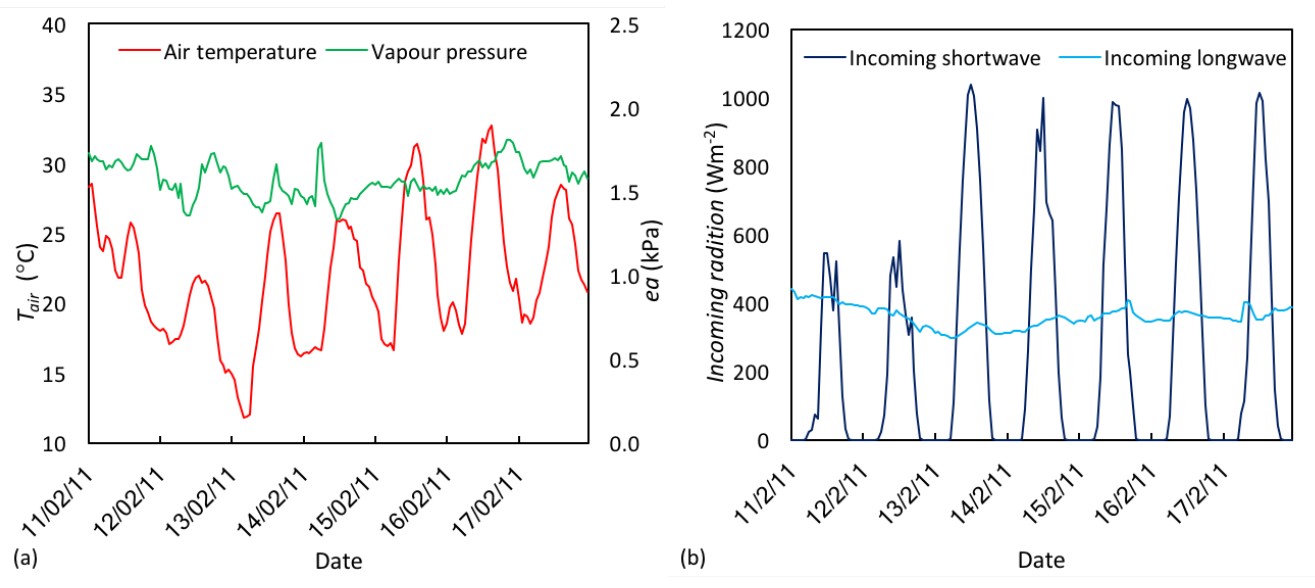

**Figure 12.** Meteorological condition during the Mawson Lakes field campaign. Data source: Bureau of Meteorology Parafield Airport (ID: 023013) and Kent Town (ID: 94675) weather stations, Adelaide.

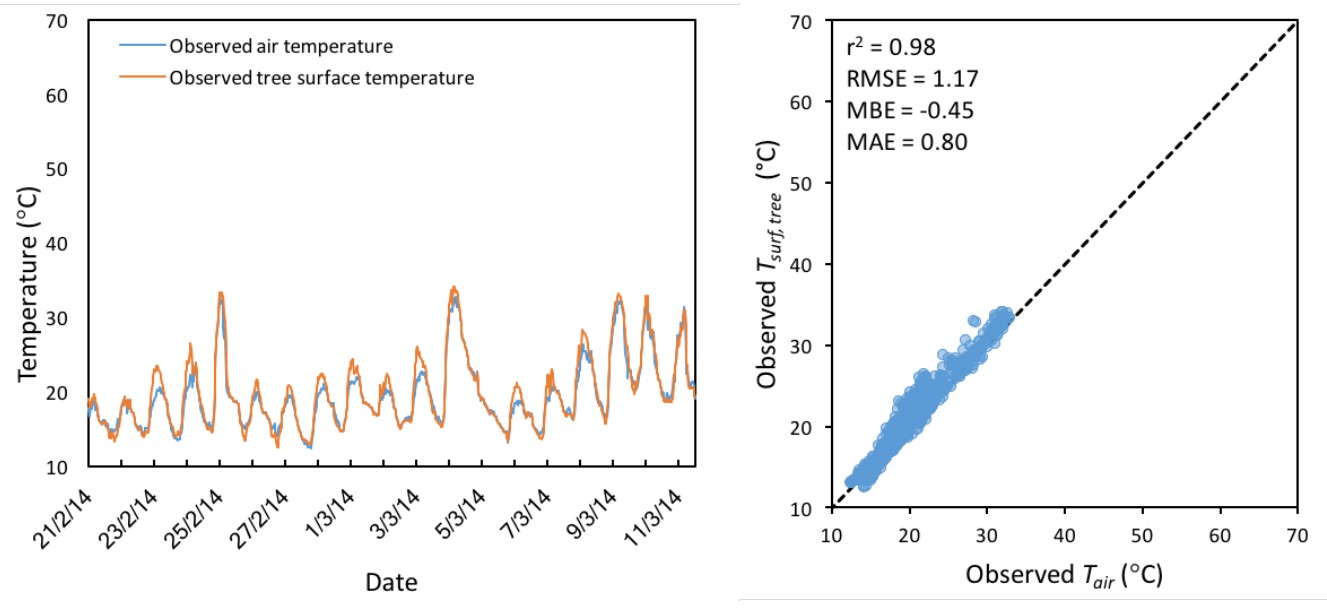

**Figure 13.** Observed air temperature vs. observed tree $T_{surf}$.