# Peer review of "The Air-temperature Response to Green/blue-infrastructure Evaluation Tool (TARGET v1.0): an efficient and user-friendly model of city cooling."

_Geoscientific Model Development, 2018_

## Referee Comment (RC1) · Anonymous Referee #1 · 2 Nov 2018

General: This paper introduces a promising approach to modelling urban temperature which could allow planners and consultants to access first-order results with little input data or computation time relative to most other models. My main reservation is that the assumptions and simplifications adopted here make the approach unsuitable for modelling spatial variations in micro-scale thermal comfort, which can vary dramatically even when air temperature variations are quite modest. This because of exposure to radiation and localized air flow, neither of which the current approach models with spatial precision.

[Figure]

Detailed: p7 lines 1-3 - "...walls and ground surfaces have similar longwave emission relative to the sky, and... solar radiation receipt can be approximated by SVF, on average. This simplification means that the model makes no distinction between lit and unlit buildings walls..." It also makes no distinction between lit and unlit ground surfaces, or pedestrians within an urban space. This should be noted as well. p12 lines 3-4 - "Utop is estimated at the top of the UCL based on Uz using a logarithmic relationship." This seems to be problematic, because the constant flux layer in which a logarithmic wind profile can be found is separated from the UCL by a Roughness Sub-Layer. Extending the logarithmic profile downward through the RSL can lead to unrealistic wind speeds. This is significant because the canyon wind speed, and in turn the surface conductances and canyon air temperature itself, are based on Utop (as described in Eqs. 16-18).

---

## Referee Comment (RC2) · Anonymous Referee #2 · 11 Dec 2018

This paper presents a climate-service-oriented tool TARGET for diagnosing near surface air temperature based on urban energy balance. The reviewer strongly agrees the motivation of the work that the accessibility of urban climate models should be improved by providing end-user-friendly tools with less demands for modelling expertise and specialized computing facilities. And the paper is well written with technical details clearly provided and results nicely presented. As such, the paper should be accepted after a minor revision.

However, the reviewer has the following concerns about this work and hope the authors can well address them:

1) Physics scheme of water surface: The choice of such a *moderately* complex lake model (Molina Martínez et al., 2006) should be justified, in particular considering OHM is used for other land surfaces, as this choice notably breaks the consistency in physics scheme for $Q_G$. Also, the lake model used in TARGET is neither simple to guarantee calculation performance (e.g., vertical discretization is required to get water temperature profiles) nor sophisticated to consider the physical rigour (e.g., band-based absorption of solar radiation is omitted). The reviewer should point out that OHM can also be used for water surfaces to obtain $Q_G$ with easy adaptation (e.g., Ward et al. (2016)).

2) Applicability for long-term applications: Although the limitation of TARGET in long-term applications has been attributed to that in OHM, it should be noted modelling advances in OHM (e.g., corrections in OHM coefficients (Ward et al., 2016), analytical determination of OHM coefficients (Sun et al., 2017)) to address this issue should be mentioned and their potential in improving TARGET can be discussed.

3) Code availability: The authors suggest the Java version of TARGET for performance reason, which interested the reviewer to review the python code in addition to the paper as more and more scientific models (e.g., Hamman et al. (2018), Monteiro et al. (2018)) are being published in Python for the easy accessibility (which TARGET claims as its key feature). After the code review, the reviewer noticed in the core calculation functions, the famous `numpy` is not well utilised to conduct heavy numerical computations. To make TARGET more accessible, the authors are very encouraged to improve the Python version for better performance and to distribute it via public repositories (e.g., PyPI).

**Minor Comments:**

1) Please justify the assumption for combined resistance between roof and canyon in equation 16. It is unclear to the reviewer how the influence of roofs can be exerted on canyons.

2) Presentation:
   a) Equation 16 is tediously long: simplify it.
   b) Section 4.1.1 --> section 4.2.
   c) $C_{watr}$ in "list of symbols" are duplicated twice.

**References:**

Hamman, J. J., Nijssen, B., Bohn, T. J., Gergel, D. R. and Mao, Y.: The Variable Infiltration Capacity model version 5 (VIC-5): infrastructure improvements for new applications and reproducibility, Geosci Model Dev, 11(8), 3481–3496, doi:10.5194/gmd-11-3481-2018, 2018.

Molina Martínez, J. M., Martínez Alvarez, V., González-Real, M. M. and Baille, A.: A simulation model for predicting hourly pan evaporation from meteorological data, J Hydrol, 318(1-4), 250–261, doi:10.1016/j.jhydrol.2005.06.016, 2006.

Monteiro, J. M., McGibbon, J. and Caballero, R.: sympl (v. 0.4.0) and climt (v. 0.15.3) – towards a flexible framework for building model hierarchies in Python, Geosci Model Dev, 11(9), 3781–3794, doi:10.5194/gmd-11-3781-2018, 2018.

Sun, T., Wang, Z.-H., Oechel, W. C. and Grimmond, C. S. B.: The Analytical Objective Hysteresis Model (AnOHM v1.0): methodology to determine bulk storage heat flux coefficients, Geosci Model Dev, 10(7), 2875–2890, doi:10.5194/gmd-10-2875-2017, 2017.

Ward, H. C., Kotthaus, S., Järvi, L. and Grimmond, C. S. B.: Surface Urban Energy and Water Balance Scheme (SUEWS): Development and evaluation at two UK sites, Urban Climate, 18, 1–32, doi:10.1016/j.uclim.2016.05.001, 2016.

---

## Author Comment (AC1) · 19 Dec 2018

General: This paper introduces a promising approach to modelling urban temperature which could allow planners and consultants to access first-order results with little input data or computation time relative to most other models. My main reservation is that the assumptions and simplifications adopted here make the approach unsuitable for modelling spatial variations in micro-scale thermal comfort, which can vary dramatically even when air temperature variations are quite modest. This because of exposure to radiation and localized airflow, neither of which the current approach models with

spatial precision.

Thank you for these comments.

The focus of the current version of TARGET (under review in this paper) is outdoor street-level air temperature, not human thermal comfort. We agree with the referee that TARGET cannot capture the micro-scale climate variations that influence human thermal comfort at the scale experienced by an individual. We will make sure that this statement is made clear in the introduction and conclusions of the amended manuscript; e.g. "TARGET calculates the average air temperature at street level in urban areas, but not does represent micro-scale variations in shading or wind flow at the human scale".

We believe that this spatially averaged approach still has great value to planners and policy-makers when evaluating blue/green infrastructure proposals.

Detailed: p7 lines 1-3 - "...walls and ground surfaces have similar longwave emission relative to the sky, and... solar radiation receipt can be approximated by SVF, on average. This simplification means that the model makes no distinction between lit and unlit buildings walls..." It also makes no distinction between lit and unlit ground surfaces, or pedestrians within an urban space. This should be noted as well. p12 lines 3-4 –

The referee is correct. We will add this comment to the amended manuscript. Thank you.

"Utop is estimated at the top of the UCL based on Uz using a logarithmic relationship." This seems to be problematic, because the constant flux layer in which a logarithmic wind profile can be found is separated from the UCL by a Roughness Sub-Layer. Extending the logarithmic profile downward through the RSL can lead to unrealistic wind speeds. This is significant because the canyon wind speed, and in turn the surface conductances and canyon air temperature itself, are based on Utop (as described in Eqs. 16-18).

Thank for this comment – we will clarify what Utop is and how it calculated in the

amended manuscript, at P12L4:

"Utop is estimated at 3xH based on the observed wind speed (Uz) at a nearby observational site (ideally an airport) using a logarithmic relationship. Airports are relatively devoid of roughness elements and wind speed is typically measured at 10 m above the surface. As such, the assumption a logarithmic profile through the roughness sublayer (Masson, 2000) is imposed."

We also note that at P6L19 we stated that: "Ideally, meteorological data should be representative of a nearby urban site" - this is incorrect - we will remove this sentence and encourage model users to take observations from a nearby airport.

—

More detailed explanation:

Utop is not estimated at the top of the UCL, but rather is an estimate of wind speed at 3x the height of the tallest building in the domain. This will be clarified in the amended manuscript. We currently use a uniform Utop value for the whole domain.

Typically, urban canopy models are forced by "above canyon" wind speed (and other meteorological variables). However, above canyon data is almost never available to model users so we wanted to devise a simple and computationally inexpensive method to diagnose Utop from observed wind speed. Hence the use of a logarithmic profile. We also note that the assumption of a log profile through the roughness sublayer is used in other urban canopy models (e.g. TEB; Masson, 2000). In TARGET (an offline model), this simplification is necessary to ensure computation efficiency – as it avoids computationally expensive iterative methods for solving the above canyon wind speed.

We worked hard when building TARGET to balance computational efficiency and complete representation of physically realistic processes. This is one area where we had to simply the represented physics to achieve a computationally efficient outcome. We will point out that this approach will be more problematic when observed wind speed

(Uz) is taken from a low measurement height (e.g. below 10 m) and/or from a site with a high roughness length (e.g. > 0.5 m) in the limitations section.

Improvement of the Utop diagnoses, while maintaining computation efficiency, could be explored in future work.
* * *

---

## Author Comment (AC2) · 19 Dec 2018

This paper presents a climate-service-oriented tool TARGET for diagnosing near surface air temperature based on urban energy balance. The reviewer strongly agrees the motivation of the work that the accessibility of urban climate models should be improved by providing end user-friendly tools with less demands for modelling expertise and specialized computing facilities. And the paper is well written with technical details clearly provided and results nicely presented. As such, the paper should be accepted after a minor revision. However, the reviewer has the following concerns about this

work and hope the authors can well address them:

We thank the referee for the constructive and thoughtful comments and agree there is a strong need for models like TARGET. We will respond to each comment below. In some cases, the original reviewer comments are broken down into sub-comments for clarity.

1a) Physics scheme of water surface: The choice of such a moderately complex lake model (Molina Martínez et al., 2006) should be justified, in particular considering OHM is used for other land surfaces, as this choice notably breaks the consistency in physics scheme for Qg.

Indeed, we wanted to use the OHM model (with force-restore) for all surfaces but we can not obtain good surface temperature results for water using OHM. We tested the parameters/modifications used in Ward et al. 2016 and still found substantial over-predictions of surface water temperature (over 10 °C) during the day.

We note that Ward et al. 2016 does not evaluate OHM performance for a 100% water surface and therefore does not truly demonstrate good model performance for water. Is the referee aware of such an evaluation in the literature?

If the OHM method in conjunction with force-restore can be shown to provide reliable estimates of water surface temperatures then we will implement that scheme in future model development.

As TARGET is a climate-service-oriented tool, we think that good model performance is more important than the consistency of physics schemes used. As such, we believe it is best to retain the lake model currently described in TARGET v1.0. We acknowledge in the manuscript the inconsistency of physics schemes used (P22L23). We will clarify, at the beginning of Section 2.6, the reasons we chose the lake model, and in the limitations (Section 6) we will re-emphasize the inconsistency with OHM.

1b) Also, the lake model used in TARGET is neither simple to guarantee calculation

performance (e.g., vertical discretization is required to get water temperature profiles) nor sophisticated to consider the physical rigour (e.g., band-based absorption of solar radiation is omitted).

We respectfully disagree with the referee here. We believe the lake model is simple enough to "guarantee calculation performance" - we find minimal differences in model speed with the lake model switched on vs. off. Most of the surfaces are treated as 2 layers in TARGET - whereas the water surface is effectively treated as 4 layers - this does not substantially impact computation time.

However, the referee is correct, the model does neglect some physical process in water bodies. This is required to ensure model efficiency and simplicity. A judgment call was made to exclude some processes associated with water, including those mentioned by the referee. Extensive testing has found no unexpected behavior due to these omissions.

We will state all the physical processes currently omitted by the model in the paper and describe any known associated limitations. If the referees knows of anything specific that we have not mentioned in the manuscript please advise. We will happily include an acknowledgment of, and reference to, any associated limitations.

1c) The reviewer should point out that OHM can also be used for water surfaces to obtain Qg with easy adaptation (e.g., Ward et al. (2016)).

We do not believe (as mentioned above) that Ward et al. (2016) demonstrates that the OHM can be used accurately for water surfaces. Is the referee aware of a paper that demonstrates accurate Qg performance for a true water surface (i.e. 100% water)? Please advise.

2) Applicability for long-term applications: Although the limitation of TARGET in long-term applications has been attributed to that in OHM, it should be noted modelling advances in OHM (e.g., corrections in OHM coefficients (Ward et al., 2016), analytical

determination of OHM coefficients (Sun et al., 2017)) to address this issue should be mentioned and their potential in improving TARGET can be discussed.

We appreciate the reviewer pointing out these exciting opportunities for potential future work. If we can develop a version of TARGET that can be reliably used for long-term simulations that would be excellent.

For now, we do not want to encourage users to conduct long-term simulations with TARGET. First, we will need to do thorough testing and evaluations with long-term datasets. However, the additions the referee has mentioned will certainly be explored. We will note this as future work in the amended paper, and cite the papers that may be used for future improvements.

3) Code availability: The authors suggest the Java version of TARGET for performance reason, which interested the reviewer to review the python code in addition to the paper as more and more scientific models (e.g., Hamman et al. (2018), Monteiro et al. (2018)) are being published in Python for the easy accessibility (which TARGET claims as its key feature). After the code review, the reviewer noticed in the core calculation functions, the famous numpy is not well utilised to conduct heavy numerical computations. To make TARGET more accessible, the authors are very encouraged to improve the Python version for better performance and to distribute it via public repositories (e.g., PyPI).

Our intention is to offer versions of the code in both Python and Java. While we agree that the use of Java is less common and perhaps less familiar in model building, we do not believe that a Java version is less accessible than one in Python. Java, after all, is currently the most widely used computer language.

Java only requires the end user to have a Java runtime installed. All of the model's external dependencies and model's source code will be precompiled and packaged into a single Jar file. Python is not always as simple, requiring the correct Python (2 or 3) to be installed, with the external dependencies (i.e. NumPy, Netcdf, etc.), often

requiring Anaconda to ensure the different versions do not conflict with other Python installations.

In our benchmarks, the Java version ran anywhere from 10-100x faster than the Python version. We agree that better use of NumPy would help optimize our Python code, as it makes array access more efficient and offers other performance gains, but we have no reason to believe based on our experimentation and research into the two languages that any optimized Python version will ever reach the performance of out of the box Java and its just in time (JIT) compiler. Achieving that magnitude of performance increase would likely require the use of tools such as Numba or Cython, requiring a level of user technical expertise that goes against our overall design principal of providing a model simple enough for anybody to use. Having said that, the performance of both versions fits well within our design goal of a modelling tool that is quick and efficient enough for widespread use. Both run quickly, just the Java version currently runs much faster.

Optimization and improvements of both versions of the model will always be an on-going goal. Once the article has been published and people start using the Github repositories, feedback or code enhancements will be encouraged and integrated into future versions. Any specific suggestion on how to improve numerical performance with Python would be welcomed.

Ongoing improvement of the Python and Java code will be carried out but this should not, in our opinion, delay the publication of the TARGET paper.

Minor Comments:

1) Please justify the assumption for combined resistance between roof and canyon in equation 16.

It is unclear to the reviewer how the influence of roofs can be exerted on canyons.

Thank you for the opportunity to clarify.

There is no consensus on how roofs might affect air temperature in the canyon. In

reality, this interaction is dependent on building height, street width, upwind configuration, presence or absence of tall trees, wind speed and atmospheric stability, to name a few factors. Some urban canopy models assume roofs can be directly connected to the canyon air (e.g. Community land model urban [Oleson et al., 2010]) while others assume no direct interaction (e.g.single-layer urban canopy model [Kusaka, 2001] or TEB [Masson 2000]).

In a coupled urban canopy model, the interaction between the roof and the urban canopy occurs indirectly via the atmospheric model. However, TARGET does not represent a two-way-interaction between the roof surfaces and the above canyon air temperature (Tb). As such, without some direct connection between the roof and canyon, there would be no way for rooftop cooling interventions (e.g. cool/green roofs) to affect canyon air temperature in TARGET.

We will add these sentences to the amended manuscript: "We hypothesized that the heat transfer from roofs to the canyon air could be approximated by two resistances in series (the canyon-to-atmosphere resistance (ca) and surface to canyon resistance (cs)). The logic here is that resistance to heat transfer from the roof surface to the canyon should be greater than ca or cs independently. Through sensitivity testing, we were able to demonstrate that this assumption improves predicted canyon air temperature."

For most cases, we think this assumption is quite reasonable (especially given other urban canopy models assume the roof can directly interact with the canyon air via a single resistance). The impact of rooftops on near-surface air temperature becomes more uncertain for taller buildings. The uncertainties here point to the need for experimental fluid dynamics work to better ascertain these resistances, and TARGET can be improved as the theory develops.

2) Presentation:

a) Equation 16 is tediously long: simplify it. We will include a simplified version of this

Interactive
comment

eq. In the amended m/s. Most likely a summation over "i"

b) Section 4.1.1 –> section 4.2. Amended

c) Cwatr' in "list of symbols" are duplicated twice. Amended

———————————————

---

## Referee Comment (RC3) · Anonymous Referee #3 · 20 Dec 2018

This study presents a simple urban climate numerical model aimed at being used as decision support tool by urban planners. The paper first presents the principles and equations of the model, then an evaluation of simulated surface temperatures and air temperatures against remote-sensed observations and in situ measurements, and finally an example of application for urban planning scenarios evaluation.

The model is intended to by applied for evaluating urban design choices at very fine scale but is however based on very simple approaches:

[Figure]

(1) The concept of urban canyon used in TARGET (without considering various building heights, street directions, street intersections, public spaces like squares etc.) is no more realistic for such spatial resolution.

(2) Some of the parameterizations are based on many simplifying assumptions, e.g.:

»Radiation calculation: it does not account for diffuse/direct partitioning of incoming radiation and applies sky-view factor approach, nor multiple radiation reflections inside the canyon. The calculation for tree canopy are not detailed so that it is not clear if the radiation transmission through foliage canopy is considered etc.

» Storage heat flux: it is calculated following an empirical formulation with constant coefficients (Eq. 5). It is not clear how they are prescribed (despite biblio references), and how they could make possible to represent the spatial heterogeneity of urban material properties.

(3) To run the surface model in offline mode, i.e. without retro action of surface processes on the low-level atmospheric conditions and without horizontal advection effect, is also a strong limitation. The spatial extend of cooling effects of green or blue infrastructures cannot be correctly captured.

The evaluation of TARGET surface temperatures on the first experimental site is good. But there is very little details about how this evaluation is done and what experimental data are used. The evaluation for the second site shows important biases of the model both for surface temperature and air temperature. This clearly highlights the limitations of the model to accurately simulate the urban climate at such a fine scale, and especially to reproduce the spatial variability of microclimate depending on urban landscape heterogeneity. The comparison to fixed stations data for air temperature shows important biases with an overestimation of air temperature in built-up environments and an underestimation in vegetated environments. One can then expect an important overestimation of the cooling effect of green infrastructures in case of greening scenarios evaluation.

In conclusion, the simplicity of the numerical tool makes it not suitable for microscale urban climate modelling, and for an accurate evaluation of urban design strategies. In the light of this finding, I do not recommend the publication of this paper.

—————————————————————

---

## Author Comment (AC3) · 30 Jan 2019

Referee #3

This study presents a simple urban climate numerical model aimed at being used as decision support tool by urban planners. The paper first presents the principles and equations of the model, then an evaluation of simulated surface temperatures and air temperatures against remote-sensed observations and in situ measurements, and finally an example of application for urban planning scenarios evaluation. The model

is intended to by applied for evaluating urban design choices at very fine scale but is however based on very simple approaches:

Thank you for your comments. We welcome the opportunity to clarify TARGET's scientific value.

Based on the comments from this referee it is clear that we need to improve communication of TARGET's purpose and limitations in the manuscript. We have also decided that based on all the referees comments we will recommend a minimum spatial resolution of 100 m. for air temperature simulations This way, the model is not attempting to resolve microscale features. We have redone our analysis at 100 m and have clarified this in manuscript (specific modifications are listed below). We hope that this adjustment will assuage the referees main concerns.

Figures 6,7,8,9,10 and Table 2 have been amended to reflect 100 m resolution simulations rather than the previously used 30 m resolution. Modifying the resolution did not substantially impact the results of the model evaluation of air temperature or the heat mitigation scenario simulations

We also updated the calculation of cooling sensitivity (equation 19) for clarity.

Before getting to some specific issues mentioned by the referee we will make some broader comments here: TARGET is not a microscale model like ENVI-met or TUF3D or CFD approaches - it cannot (and is not designed to) capture micro-scale climate variations that influence human thermal comfort at the scale experienced by an individual. It is designed to be used at the "canyon" to "block" scale - these features fall in the overlap between "micro-" and "local-scales" according to the commonly used nomenclature in urban climatology (see Figure below).

Figure: Time and horizontal space scales of selected urban climate dynamics and wind phenomena (Oke et al., 2017).

The issues discussed above were raised by the other reviewers, and we have added a

clarification to the introduction as follows (at P3L23):

"TARGET calculates the average air temperature at street level in urban areas, but does not represent micro-scale variations of radiation exchange or wind flow at the human scale. The model is designed to be used at the urban canyon-to-block scales (100 - 500 m). We recommend a minimum spatial resolution of 100 m for air temperature simulations and 30 m for surface temperature. It can be used to assess the canyon averaged impacts of street scale interventions or larger-scale suburban greening projects. TARGET is climate-service-oriented tool that provides a first order approximation of the impacts of GBI on surface temperature and street level air temperature to provide scientific guidance to practitioners during the planning process."

We will remove the following sentence (P3 L25): "TARGET is formulated to be applied at the micro-to-local-scales (street-to-precinct scales); meaning it can be used to assess the cooling benefits of small scale interventions (e.g. a single street or small urban park) to suburb scale greening projects".

Apart from the communication in the original manuscript, we sense that the referee disagrees philosophically with the approach taken. The authors of this paper (and indeed the other 2 reviewers) believe that there is scientific value in a simple and accessible model that can generate first order estimates of local cooling impacts. Such a model will limit complexity wherever it is unnecessary and use simplified physical representations where possible. A fit-for-purpose model balances the level of physical representation, computational efficiency and ease of use appropriately. We believe we have done so here. Relative to more complex urban climate models with more complete physical representations, we use simplified physics in favour of a computationally- and parameter-light and user-friendly model. All models are abstractions of reality. The role of a model developer is to choose the appropriate type and degree of abstraction for the purpose at hand.

There are already many complex urban climate models that can be used by trained

scientists with access to powerful computers. However, these complex models are, by and large, not used by practitioners or environmental consultants who work with policy-makers. Therefore, TARGET's simplicity is by design; motivated by the need for such tools in the planning and policy community. We believe TARGET should be judged with the goals of computational efficiency and accessibility in mind.

We posit that, in many ways, building a simple but reliable model is more difficult than a complex model that includes relatively complete, computationally-intensive physical representation of processes. We believe strongly that a carefully designed and robustly evaluated simple model like TARGET does represent a valuable scientific contribution.

We respond to each comment below.

(1) The concept of urban canyon used in TARGET (without considering various building heights, street directions, street intersections, public spaces like squares etc.) is no more realistic for such spatial resolution.

The canyon approach utilized by TARGET is widely used at the neighbourhood scale: Krayenhoff & Voogt (2010); Yang & Wang (2015); Song & Wang (2015); Broadbent et al., (2018).

More specifically, the reviewer mentions 4 features: "building heights, street directions, street intersections, public spaces like squares".

Street orientations: The author is correct to point out that TARGET cannot be used to assess the impacts of different street orientations. Street orientation has a substantial impact on wall temperatures and mean radiant temperature (MRT) (Johansson, 2006), but the effect on air temperature is a 2nd order impact, due to atmospheric mixing. This is especially true for relatively low density urban areas/suburban areas (open low rise LCZ5 and LCZ6), which make up the largest proportion of cities (See Figure below from Matthias Demuzere).

We do not believe that the additional data requirements and preprocessing needed to

include street orientation is justified for a first order approximation of street level air temperature.

Figure: Global distribution of local climate zones (LCZs) (provided by Matthias Demuzere).

Building heights:

Building height is in fact accounted for. Heterogeneous building heights are included (model resolution average) - these heights ultimately influence the radiation received by the canyon and the heat exchange coefficients. However, it is true to say that TARGET cannot represent sub-grid scale building height heterogeneity. For example, a single tall building amongst lower buildings is not directly accounted for.

Street intersections, public spaces:

Again, these features cannot be explicitly resolved by TARGET (i.e the exact geometry). A intersection or plaza will be represented as a "canyon" without any walls. We believe that the first order impacts of these features are captured by TARGET. Only microscale models (e.g., ENVI-met, TUF3D, CFD models) captures these geometrical features, but those models are orders of magnitude more computationally intensive than TARGET.

(2) Some of the parameterizations are based on many simplifying assumptions, e.g.: Âż Radiation calculation: it does not account for diffuse/direct partitioning of incoming radiation and applies sky-view factor approach, nor multiple radiation reflections inside the canyon.

The author is correct we do not account for direct / diffuse partitioning of incoming shortwave radiation. We could add this feature to the model but we do not believe it would add much to the air temperature calculation. However, these physics will be more important for future work in human thermal comfort realm. Adding direct/diffuse partitioning increases data input requirements and given our guiding principles of sim-
plicity and low input data requirements it does not seem justified.

The referee is also correct that multiple reflections are not considered.

The calculation for tree canopy are not detailed so that it is not clear if the radiation transmission through foliage canopy is considered etc.

Radiation transmission through tree foliage is not considered. This does needs to be clarified in the manuscript. We propose adding this at P4 L7:

"To represent the first order shading impacts of trees, we effectively represent tree canopy as part of the urban canyon. As shown in Figure 1, the width of the canyon (and therefore the amount of radiation the enters and leaves the canyon) is modulated by the planar area of trees. The simple method, implies that none of the radiation effectively "intercepted" by trees enters the canyon. The area underneath trees (not shown in planar land cover maps) is added to the model to represent the additional thermal mass. This simple approach allows for a first order representation of two major process associated with trees: solar shading and longwave trapping."

Storage heat flux: it is calculated following an empirical formulation with constant coefficients (Eq. 5). It is not clear how they are prescribed (despite biblio references), and how they could make possible to represent the spatial heterogeneity of urban material properties.

The OHM parameters are taken from the literature and are prescribed for each land cover category, as indicate by literature references indicated in Table 1.

Spatial heterogeneity:

The surface heterogeneity of surface types that can be simulated (e.g. dry grass, irrigated grass, asphalt, concrete, buildings etc), is consistent with other microclimate models e.g. ENVI-Met. The land cover categories chosen for TARGET are representative of typical categories found in urban land cover maps that practitioners usually have access to. Adding additionally land cover categories/heterogeneity would increase input data requirements and must be avoided.

(3) To run the surface model in offline mode, i.e. without retro action of surface processes on the low-level atmospheric conditions and without horizontal advection effect, is also a strong limitation. The spatial extend of cooling effects of green or blue infrastructures cannot be correctly captured.

The referee is correct that the model uses an offline method - and it is correct to state that low-level atmospheric mixing is not captured in the model. There are no low computational cost methods for representing atmospheric mixing to our knowledge.

Further, a benefit of excluding advection is that the cooling effects of green infrastructure are "entirely local" - i.e. no horizontal mixing. We would argue that this is a useful way to report cooling magnitudes as practitioners can more easily understand a "maximum local" benefit associated with design proposal. Secondly, advection is case specific - its direction and magnitude will vary throughout the day and between individual days, which is not useful for generalized (i.e., time-averaged) results of the type typically sought by planners and policy makers. Again, omission of advection functions to create a fit-for-purpose model.

This is explained at P22L4:

"For computational efficiency, the model assumes no horizontal advection (inside or above) the UCL. In general, advection reduces the local impacts (i.e. cooling directly adjacent the cooling intervention) of GBI due to atmospheric mixing, and therefore we expect TARGET to provide estimates of near maximum cooling benefits at the scale of model application. In reality, cooling effects will be diminished somewhat by advection, especially during the day and during high wind conditions."

The evaluation of TARGET surface temperatures on the first experimental site is good. But there is very little details about how this evaluation is done and what experimental data are used.

Below is what we have in manuscript on model evaluation - we have added some details in quotations marks below (P12L23):.

"To test model performance at simulating Tsurf of different land cover classes and perform sensitivity analysis on a number of model parameters, we used ground-based observations of Tsurf from the Melbourne metropolitan area. Coutts et al. (2016) deployed infrared temperature sensors (SI-121 - Apogee), during February 2012 "(5 min averages)", across a range of land cover types including: asphalt, concrete, grass, irrigated grass, steel roof, and water. "Infrared sensors were mounted above the aforementioned surface types installed at heights of approximately 1.5–2 m". The conditions during this period represented near-typical summertime conditions in Melbourne; including a number of days (15th, 24th, and 25th February) where air temperature exceeded 30 C (see Fig. 11). These hotter days were characterised by northerly winds, which bring hot and dry air from Australia's interior, and often result in heatwave conditions in Melbourne. Additionally, there was at least one cloudy day where incoming shortwave radiation (K↓) dropped significantly and negligible amount of rainfall occurred (17th February). "To compare the Coutts et al. (2016) observations with TARGET we ran the model for each surface type (i.e. 100 % grass or roof etc) with radiation forcing data from the Melbourne Airport weather station during the time period in question. The Tb calculation was not needed since we only calculated Tsurf for this part of the model evaluation. The 30 min output from TARGET was compared with Tsurf observations and statistics were calculated."

The evaluation for the second site shows important biases of the model both for surface temperature and air temperature. This clearly highlights the limitations of the model to accurately simulate the urban climate at such a fine scale, and especially to reproduce the spatial variability of microclimate depending on urban landscape heterogeneity. The comparison to fixed stations data for air temperature shows important biases with an overestimation of air temperature in built-up environments and an underestimation in vegetated environments. One can then expect an important overestimation of the

cooling effect of green infrastructures in case of greening scenarios evaluation. In conclusion, the simplicity of the numerical tool makes it not suitable for microscale urban climate modelling, and for an accurate evaluation of urban design strategies. In the light of this finding, I do not recommend the publication of this paper.

We appreciate the referee's concern, and we disagree with their assessment.

Firstly, the model biases mentioned by the referee are expected given the aforementioned lack of advection but should be clarified to (P17L11):

"The modelled air temperatures are biased towards warmer air temperature in urban areas and cooler air temperature in rural areas. These biases are partly driven by the lack of advection in the model. Without atmospheric mixing, the local impacts of pervious and impervious surfaces are exaggerated causing an additional cooling and warming effect in rural and urban areas, respectively."

Nevertheless, we disagree with the suggestion that model performance is poor, particularly when the wider context of urban model evaluation is considered. The most widely used model in urban microclimate modeling is ENVI-met.

Please find a summary table of ENVI-met studies below with comparable model biases to those reported in this study. There are many ENVI-met studies and a complete summary is not possible here - the table below includes a cross-section of studies from different locations with comparable model performance to TARGET.

Note that most of these studies evaluate their findings against a mere 1 or 2 stations. An evaluation of ENVI-met against 27 dispersed weather stations (i.e. not a transect) would not be possible given the computation demand. As such, we believe that the evaluation in this paper is robust and demonstrates acceptable model performance given the model simplicity. As such, we disagree with the implication that the model performance is poor.

REFERENCES

Johansson, E. (2006). Influence of urban geometry on outdoor thermal comfort in a hot dry climate: A study in Fez, Morocco. Building and environment, 41(10), 1326-1338.

Broadbent, A. M., Coutts, A. M., Tapper, N. J., & Demuzere, M. (2018). The cooling effect of irrigation on urban microclimate during heatwave conditions. Urban Climate, 23, 309-329.

Berardi, U. (2016). The outdoor microclimate benefits and energy saving resulting from green roofs retrofits. Energy and Buildings, 121, 217–229

Chow, W. T., & Brazel, A. J. (2012). Assessing xeriscaping as a sustainable heat island mitigation approach for a desert city. Building and Environment, 47, 170-181.

Emmanuel, R., & Fernando, H. (2007). Urban heat islands in humid and arid climates: role of urban form and thermal properties in Colombo, Sri Lanka and Phoenix, USA. Climate Research, 34, 241–251. doi:10.3354/cr00694

Emmanuel, R., & Loconsole, A. (2015). Green infrastructure as an adaptation approach to tackling urban overheating in the Glasgow Clyde Valley Region, UK. Landscape and Urban Planning, 138, 71–86 doi:10.1016/j.landurbplan.2015.02.012

Emmanuel, R., Rosenlund, H., & Johansson, E. (2007). Urban shading—a design option for the tropics? A study in Colombo, Sri Lanka. International journal of climatology, 27(14), 1995-2004.

Ghaffarianhoseini, A., Berardi, U., & Ghaffarianhoseini, A. (2015). Thermal performance characteristics of unshaded courtyards in hot and humid climates. Building and Environment, 87, 154-168.

Goldberg, V., Kurbjuhn, C., & Bernhofer, C. (2013). How relevant is urban planning for the thermal comfort of pedestrians? Numerical case studies in two districts of the City of Dresden (Saxony/Germany). Meteorologische Zeitschrift, 22(6), 739-751.

Lin, B.-S., & Lin, C.-T. (2016). Preliminary study of the influence of the spatial arrangement of urban parks on local temperature reduction. Urban Forestry & Urban Greening, 20, 348–357. doi:10.1016/j.ufug.2016.10.003

Ng, E., Chen, L., Wang, Y., & Yuan, C. (2012). A study on the cooling effects of greening in a high-density city: An experience from Hong Kong. Building and Environment, 47, 256–271. doi:10.1016/j.buildenv.2011.07.014

Salata, F., Golasi, I., Vollaro, A. de L., & Vollaro, R. de L. (2015). How high albedo and traditional buildings' materials and vegetation affect the quality of urban microclimate. A case study. Energy and Buildings, 99, 32–49. doi:10.1016/j.enbuild.2015.04.010

Yang, J., & Wang, Z.-H. (2015). Optimizing urban irrigation schemes for the trade-off between energy and water consumption. Energy and Buildings, 107, 335–344.doi:10.1016/j.enbuild.2015.08.045

Song, J., & Wang, Z. H. (2015). Impacts of mesic and xeric urban vegetation on outdoor thermal comfort and microclimate in Phoenix, AZ. Building and Environment, 94, 558-568.

Wang, Y., Bakker, F., de Groot, R., Wortche, H., & Leemans, R. (2015). Effects of urban trees on local outdoor microclimate: synthesizing field measurements by numerical modelling. Urban ecosystems, 18(4), 1305-1331.

[Figure]

Figure: Time and horizontal space scales of selected urban climate dynamics and wind phenomena (Oke et al., 2017).

**Fig. 1.**

street orientation is justified for a first order approximation of street level air temperature.

[Figure]

Figure: Global distribution of local climate zones (LCZs) (provided by Matthias Demuzere).

**Fig. 2.**

Summary of ENVI-met microscale studies

| Study | Location | Evaluation sites | Evaluation length | $R^2$ | RMSE |
|---|---|---|---|---|---|
| **Current study** | **Adelaide** | **27** | **48 h** | **$R^2$ =0.92** | **2.0 °C** |
| Berardi (2016) | Toronto | 1 | 24h | $R^2$ =0.92 | - |
| Chow & Brazel (2012) | Phoenix | 2 | 24h | $R^2$ =0.67,0.74 | 2.79 °C 2.79 °C |
| Emmanuel & Fernando (2007) | Colombo and Phoenix | 2 | 24h | - | 2.7 °C, 2.6 °C |
| Emmanuel & Loconsole (2015) | Glasgow | 1 | 24h | $R^2$ =0.95 (Slope = 0.60) | 0.83 |
| Ghaffarianhoseini et al. (2015) | Kuala Lumpur | 1 | 14h | $R^2$ =0.96 (Slope = 1.32) | - |
| Goldberg et al. (2013) | Dresden | 1 | 24h | (Max bias = -8 °C) | - |
| Lin & Lin (2016) | idealized | 2 | 10h | (Slope = 0.36, 0.57) | 1.62 °C, 1.32 °C |
| Ng et al. (2012) | Hong Kong | 1 | 12 days | $R^2$ =0.63 (Slope = 1.74) | - |
| Song & Park (2015) | Changwon City | 27 | 3 * 6 h | $R^2$=0.63,0.32, 0.61 (Slope = 1.22, 0.41, 0.14) | 4.6 °C, 3.4 °C, 6.5 °C (27 stations combined RMSE) |
| Wang et al.. (2015) | Assen | 5 | 24 h | $R^2$ =0.73-0.98 | 0.31 - 2.13 |

**Fig. 3.**

---

## Editor Decision (ED1)

**Comments:**

- last sentenace abstract: *"TARGET will be made available to the public and ongoing development, including a graphical user interface, is planned for future work."* Should'nt this be " TARGET **is available** to the public [...]" (see Code availability Section) ?

- P. 5, l.5-6: *" Ideally, meteorological data should be representative of a nearby urban site. However, the nearest airport weather station will suffice."* In the reply to referee #1 you declated this sentence to be incorrect and that it will be deletes in the revised manuscript. So why is it still present?

- P.12 L. 7-9: *"Utop is estimated at the top of the UCL based on Uz using a logarithmic relationship. Utop is estimated at 3H based on the observed wind speed at a nearby observational site (ideally an airport) using a logarithmic relationship."* I do not think both sentences are valid???

- Chapter 6: In your reply to referee #2 you wrote "We clarify, at the beginning of Section 2.6, the reasons we chose the lake model, and in the limitations (Section 6) we will re-emphasize the inconsistency with OHM."
  In Sect. 2.6 you write *"The simple water body model is used because the OHM-force-restore method can not be reliably applied to water surfaces. "* Which leaves the reader to guess why this is the case. Could you add an explanation or a citation to support this statement?
  Furthermore, according to your replys to referees #1 and #3, I did not expect to read the word "microscale" so often.

**Typos:**

- P. 4, l. 9 "the enters" → "that enters"

- P.22 , l. 6 "[...] horizontal advection (inside or above) the UCL. [...] " remove brackets, as the sentence is not a full sentence without the content of the bracket.

---

## Author Response (AR2)

Thank you for the comments - we have addressed all the comments below. we also made some minor editorial changes. All changes are listed below and I have provided a mark-up version of the manuscript for your convenience.

Comments:

- last sentenace abstract: "TARGET will be made available to the public and ongoing development, including a graphical user interface, is planned for future work." Should'nt this be " TARGET is available to the public [...]" (see Code availability Section) ?

Yes - amended.

- P. 5, l.5-6: " Ideally, meteorological data should be representative of a nearby urban site. However, the nearest airport weather station will suffice." In the reply to referee #1 you declated this sentence to be incorrect and that it will be deletes in the revised manuscript. So why is it still present?

This has been amended to (P5L9-10):

"Meteorological data should taken from a nearby airport or an open site with minimal buildings"

- P.12 L. 7-9: "Utop is estimated at the top of the UCL based on Uz using a logarithmic relationship. Utop is estimated at 3H based on the observed wind speed at a nearby observational site (ideally an airport) using a logarithmic relationship." I do not think both sentences are valid???

Yes, the first sentence was supposed to be removed.

It now reads (P11 L25): "Utop is estimated at 3H based on the observed wind speed at a nearby observational site (ideally an airport) using a logarithmic relationship."

- Chapter 6: In your reply to referee #2 you wrote "We clarify, at the beginning of Section 2.6, the reasons we chose the lake model, and in the limitations (Section 6) we will re-emphasize the inconsistency with OHM."
- In Sect. 2.6 you write "The simple water body model is used because the OHM-force-restore method can not be reliably applied to water surfaces." Which leaves

the reader to guess why this is the case. Could you add an explanation or a citation to support this statement?

I do not have a citation as I do not know of a case where OHM has been evaluated against a "water only" surface. Our own testing is the reason that we state that the "OHM-force-restore method cannot be reliably applied to water surfaces".

I propose that we change the statement to:

P9 L26-30:

"Our analysis suggests that the OHM-force-restore method cannot be used to reliably reproduce water surface temperatures.  We tested the parameters/modifications used by Ward et al. (2016) and found substantial over-predictions of surface water temperature (over 10 °C) during the day. As such, we developed a simple water body model to stand in for the OHM-force-restore method."

- Furthermore, according to your replys to referees #1 and #3, I did not expect to read the word "microscale" so often.

We have gone through the manuscript and removed a number of mentions of the word "microscale"  or "finescale" - please see the mark-up version of the pdf to see changes:

Abstract L8-9

P5 L6

P13 L5

P19 L12

P2 1L2

P22 L11

- Typos
- P. 4, l. 9 "the enters" → "that enters"

Amended.

- P.22,l.6 "[...]horizontal advection(insideorabove)theUCL.[...]" remove brackets, as the sentence is not a full sentence without the content of the bracket.

[Figure]

Additionally, minor editorial amendments were made at (see mark-up):

P3 L8

P3 L23

P3 L28

P12 L7-8

P16 L9

P18 L20-24

P18 L28

P19 L4

[revised manuscript text omitted]